# Analyzing Molecular Determinants of Nanodrugs’ Cytotoxic Effects

**DOI:** 10.3390/ijms26146687

**Published:** 2025-07-11

**Authors:** Alicia Calé, Petra Elblová, Hana Andělová, Mariia Lunova, Oleg Lunov

**Affiliations:** 1FZU—Institute of Physics of the Czech Academy of Sciences, 182 21 Prague, Czech Republic; cale@fzu.cz (A.C.); elblova@fzu.cz (P.E.); andelovah@fzu.cz (H.A.); mariialunova@googlemail.com (M.L.); 2Faculty of Mathematics and Physics, Charles University, 121 16 Prague, Czech Republic; 3Institute for Clinical & Experimental Medicine (IKEM), 140 21 Prague, Czech Republic

**Keywords:** cytotoxicity, nanodrugs, nanoparticles, ROS, oxidative stress, molecular mechanisms of nanotoxicity

## Abstract

Nanodrugs hold great promise for targeted therapies, but their potential for cytotoxicity remains a major area of concern, threatening both patient safety and clinical translation. In this systematic review, we conducted a systematic investigation of nanotoxicity studies—identified through an AI-assisted screening procedure using Scopus, PubMed, and Elicit AI—to establish the molecular determinants of nanodrug-induced cytotoxicity. Our findings reveal three dominant and linked mechanisms that consistently act in a range of nanomaterials: oxidative stress, inflammatory signaling, and lysosomal disruption. Key nanomaterial properties like chemical structure, size, shape, surface charge, tendency to aggregate, and biocorona formation control these pathways, modulating cellular uptake, reactive oxygen species generation, cytokine release, and subcellular injury. Notably, the most frequent mechanism was oxidative stress, which often initiated downstream inflammatory and apoptotic signaling. By linking these toxicity pathways with particular nanoparticle characteristics, our review presents necessary guidelines for safer, more biocompatible nanodrug formulation design. This extensive framework acknowledges the imperative necessity for mechanistic toxicity assessment in nanopharmaceutical design and underscores the strength of AI tools in driving systematic toxicology studies.

## 1. Introduction

Nanomedicine holds enormous promise to transform today’s healthcare through its potential to enable targeted therapy, specific diagnostics, and smart drug delivery systems [1,2]. According to the National Institutes of Health, nanomedicine is a branch of nanotechnology designed for highly targeted medical intervention at the molecular level either to cure disease or repair damaged tissue [3]. Using materials and mechanisms on the nanoscale, nanomedicine offers new solutions to complex medical challenges. One of the basic advantages of nanomedicine is the ability to simplify the delivery of therapeutic agents in order to improve their therapeutic activity and minimize undesired side effects [1,2,4,5]. Many active pharmaceutical ingredients (APIs) have poor solubility, decreasing their therapeutic effectiveness and increasing the risk of undesired side reactions [6]. Nanotechnology overcomes this problem through improving drug solubility by encapsulation in lipid or polymer-based carriers, which are commonly referred to as nanocarriers [1,2,4,5]. These systems enhance the bioavailability of drugs, reduce the required dose, lower treatment costs, and reduce toxicity [1,2,4,5,6]. Additionally, nanocarriers also facilitate targeted delivery. By functionalizing the surfaces of nanoparticles (NPs) with targeting ligands, one can target nanodrugs to particular cell types, such as cancer or immune cells. This targeted delivery not only enhances therapeutic specificity but also reduces off-target effects, thereby enhancing patient outcomes [4,7,8].

Given these advantages, engineered nanoparticles have attracted growing attention in biomedical studies and clinical translation. As of 2022, the U.S. Food and Drug Administration (FDA) had approved 23 nanoparticle-based drugs, with many more under investigation in clinical trials [8,9,10]. Most of the approved nanomedicines are founded on developed platforms such as liposomes, micelles, polymeric nanoparticles, and inorganic particles [6,7,8]. Notwithstanding the impressive clinical value demonstrated by these systems, they are not without limitations.

Despite heavy investment and progress—driven in part by initiatives like the National Nanotechnology Initiative—the number of clinically approved nanomedicines is far lower than initially anticipated [6]. One major obstacle is the translational gap between preclinical (especially animal-based) studies and successful human applications [11,12,13]. This gap occurs primarily due to the absence of complete understanding of interspecies variances in physiological and pathological responses, which directly affect how nanomedicines behave in the human body [14]. Some studies state that current preclinical models overestimate treatment effects as they fail to replicate human pathophysiology properly [15]. Besides species differences, patient-to-patient variation (i.e., genetic variation, disease status, and immune status) also prevents clinical translation [6]. Other technical challenges include nanoparticle size and shape heterogeneity, instability under physiological conditions, and long-term safety and cytotoxicity issues [5,10,16,17,18,19]. Variations in nanoparticle properties can undermine reproducibility and therapeutic effectiveness, while chemical and physical instability can impact shelf life and bioactivity. In addition, with the development of nanomedicine comes growing concern over potential toxicological implications, demanding ever greater rigorous investigation of nanoparticle safety [10,16,17,18,19]. Finally, regulatory and clinical framework weaknesses are due to improper biosafety evaluations, poor patient stratification, and the absence of nanomedicine-specific guidelines [20]. Together, the findings explain the complex mechanisms responsible for the absence of correlation between successful animal trials and successful human applications in the progress of nanomedicine [21]. These limitations have had practical consequences. Safety issues and clinical failures have led to the market withdrawal of several nanodrugs [10,19,22,23,24,25,26]. In fact, FDA-approved nanomedicines dropped from over 50 in 2016 to just 23 in 2022 [9,10,27,28]. This decline underscores the pressing need for additional innovation in nanoparticle design, manufacturing scale-up, and safety testing.

One of the most important barriers to broader clinical application is the potential cytotoxicity of nanodrugs [9,10,27,28]. While adverse drug reactions, along with toxicity, are not unique to nanomedicines (they are also prevalent for conventional therapies [29,30,31]) the unique properties of nanoscale materials present some new safety concerns. An understanding of drug toxicity on the cellular and molecular levels is required for the development of safer formulations, the identification of predictive biomarkers, and the reduction in late-stage clinical failures [32]. Nanomaterials can cross cell membranes, biological barriers, and tissue interfaces more easily compared to materials at the larger size scale [6,33]. Extensive biokinetic research, however, suggests that particle size alone may have little influence on overall biodistribution [34]. Further basic understanding of complex interactions between nanomaterials and human biological systems is thus a priority for maximizing clinical efficacy and guaranteeing safety. Although a growing number of reviews cover the general toxicity and safety of nanomaterials [6,35,36,37], there have been relatively few that have systematically discussed cellular and molecular mechanisms of cytotoxicity for clinically approved nanodrugs [2,10]. This review strives to fill this gap by presenting a critical overview of molecular determinants of nanodrug-induced cytotoxicity. By doing so, we aim to gain valuable insights into the determinants of their safety profiles and inform the design of next-generation nanotherapeutics.

## 2. Materials and Methods

In order to systematically assess the current literature on nanodrug-induced cytotoxicity, we employed a multi-step search strategy combining traditional database searching with AI-driven tools, as well as manual reference list screening. The principal literature search was conducted using the Scopus and PubMed databases. To achieve greater coverage, we manually screened eligible articles’ reference lists, thus allowing us to include supplementary relevant studies that could have otherwise been missed in light of varying keyword indexing. To supplement the depth and efficiency of our review, we incorporated Elicit AI (https://elicit.com, accessed on 5 May 2025), an artificial intelligence-based research assistant built on top of the Semantic Scholar corpus (https://www.semanticscholar.org/, accessed on 5 May 2025), which includes over 126 million scholarly publications. Elicit AI was specifically programmed to identify research on nanodrug-induced cytotoxicity. This yielded 498 papers initially, which were subsequently screened based on predetermined inclusion criteria. A complete list of these studies is given in the Appendix A (Appendix A).

The entire selection, screening, and data extraction workflow is summarized in Appendix A (Appendix A), which presents a flowchart constructed in accordance with systematic reviews guidelines. The figure distinctly describes each phase of the review process—ranging from initial identification of records to final inclusion—marking methodological rigor and reproducibility. We applied the following inclusion criteria to screen articles retrieved by Elicit AI:Investigated FDA-approved or clinically used nanodrugs in therapeutic uses.Addressed molecular mechanisms of cytotoxicity induced by nanodrugs (e.g., protein corona formation, oxidative stress, inflammation, lysosomal damage).Provided experimental evidence (in vitro or in vivo) and not solely theoretical or computational.Provided a comprehensive toxicity or safety profile, such as dose–response curves or IC_50_ values.Analyzed nanodrug metabolism, degradation, or biodistribution in biological systems.Focused on therapeutic applications, excluding environmental or non-medical studies.Were original research articles or systematic reviews/meta-analyses.Addressed molecular-level toxicity mechanisms, beyond drug delivery efficacy.

Each publication was evaluated holistically against these criteria, with inclusion decisions based on overall relevance and scientific merit. To prioritize studies for in-depth analysis, Elicit AI was used to assign a relevance score to each publication. These scores, presented in Appendix A (Appendix A), reflect how well each study aligned with the selection criteria. Applying a cutoff score of ≥3.5, we identified 98 high-priority articles for detailed evaluation.

For all these 98 studies, Elicit AI facilitated the quantitative and qualitative data extraction on nanomaterial cytotoxicity. Data extracted were:Toxicity rates or IC_50_ values.Dose-dependent toxicity responses.Molecular mechanisms (e.g., oxidative stress, inflammatory signaling, lysosomal disruption).Affected cellular structures (e.g., mitochondria, lysosomes, membrane integrity).Biomarkers of oxidative stress and inflammation.Observed immunomodulatory effects.

In reports describing several pathways of toxicity, Elicit AI was instructed to order mechanisms by reported rank or frequency of citation. If no explicit molecular mechanisms were reported, the report was labeled as “No specific molecular mechanisms reported.”

To ensure consistent and targeted data extraction, Elicit AI was guided using the following screening questions:

*Therapeutic Nanodrug Focus*: Does the study investigate FDA-approved or clinically used nanodrugs and their cytotoxic effects at the molecular level?

*Molecular Interaction Analysis*: Are molecular-scale interactions (e.g., protein corona formation) between nanomaterials and cells examined?

*Experimental Validation*: Is the research based on empirical (in vivo or in vitro) data rather than computational or theoretical models?

*Safety Assessment*: Is a comprehensive toxicity or safety profile provided?

*Biological Fate*: Does the study examine nanodrug degradation, metabolism, or biodistribution in biological systems?

*Study Type Relevance*: Is the context therapeutic (medical) rather than environmental or non-clinical?

*Evidence Synthesis*: Is the publication an original research article or a systematic review/meta-analysis?

*Mechanism Analysis*: Does the study focus on molecular mechanisms of cytotoxicity rather than drug delivery efficiency alone?

This structured, AI-assisted methodology ensured a robust, unbiased, and reproducible selection and analysis of literature addressing nanodrug-induced cytotoxicity.

During manuscript preparation, ChatGPT 4o (OpenAI) was used for grammar refinement, academic style enhancement, English translation, and consistency checks. All content generated or edited with AI assistance was thoroughly reviewed and revised by the authors. The authors accept full responsibility for the scientific content, interpretation, and integrity of the work. It is important to note that AI tools, including Elicit AI Pro and ChatGPT 4o, do not meet the criteria for authorship and are therefore not listed as co-authors. The authors maintain full accountability for the originality, accuracy, and scholarly validity of this manuscript.

It is important to acknowledge several limitations of this study. Primarily, the analysis is based on a qualitative synthesis of cytotoxicity data, which restricts the ability to identify consistent toxicity trends or establish clear dose–response relationships across studies. Our objective, however, was to provide a broad overview of the cytotoxicity associated with nanodrugs and to highlight common molecular mechanisms underlying these effects. Furthermore, the study incorporates findings from both in vitro and in vivo models, which vary in their physiological relevance to human biology. While this approach offers a broader perspective, it also introduces variability that may limit the generalizability of the findings. In addition, the study intentionally excludes data related to environmental or non-medical nanotoxicity—such as inhalation risks from industrial nanomaterials. While this may omit certain mechanistic insights relevant to human exposure in broader contexts, the exclusion was deliberate in order to maintain a focused analysis on the pharmaceutical applications of nanomaterials.

## 3. Types of Nanodrugs Approved for Clinical Applications

Prior to providing an overview of molecular mechanisms involved in the possible nanodrug-induced cytotoxicity of nanodrugs used in the clinic, let us first define a “nanodrug” and provide an overview of different classes of existing nanodrugs utilized for clinical use. We start with definitions of the key terms “nanoparticle,” “nanomedicine,” and “nanodrug.” A review of the literature indicates that some of these terms are differently defined in the scientific community and the regulatory arena [10,38,39]. In the majority of the scientific literature, nanoparticles are mainly characterized based on their size and related physicochemical characteristics [10,38,39]. While some (mostly referring to regulatory documents) consider nanoparticles to be 1–100 nanometers in size, others consider the size limit up to 1000 nanometers [10,38,39]. Notably, the European Medicines Agency (EMA) considers the size range of nanoparticles as 0.2–100 nm [40]. Conversely, the term nanomedicine is broadly defined in terms of function, rather than dimension [39,41,42]. It is utilized to refer to the application of nanotechnology for diagnosing disease, treating disease, and preventing disease. The definition of nanomedicine also depends on context. In certain contexts, it includes the therapeutic application of nanomaterials—including diagnostic, therapeutic, and monitoring applications [39]. In more general contexts, it is referred to as a multi-disciplinary science that utilizes nanotechnology to enable new imaging, diagnosis, treatment, tissue repair, and regeneration of biological systems [41]. Likewise, the term nanodrug is not standardized. Most commonly, it is used to describe a drug product that utilizes nanoscale materials (i.e., nanoparticles or nanocarriers) to improve drug delivery, efficacy, and safety [10,38,39].

### 3.1. Defining the Terminology of Nanodrugs

It is interesting to point out here that the FDA has not adopted official regulatory definitions of the terms “nanotechnology,” “nanomaterial,” or “nanoscale” [43]. The FDA did suggest, however, a working definition for products involving the application of nanotechnology. The product uses nanotechnology, according to the agency, if it matches two criteria:“a material or end product is engineered to have at least one external dimension, or an internal or surface structure, in the nanoscale range (approximately 1 nm to 100 nm);a material or end product is engineered to exhibit properties or phenomena, including physical or chemical properties or biological effects, that are attributable to its dimension (s), even if these dimensions fall outside the nanoscale range, up to one micrometer (1000 nm)” [43].

This definition highlights both structural properties and nanoscale-dependent functional properties, with flexibility to accommodate emerging nanotechnologies in pharmaceutical and biomedical applications. The following are the implications of this definition. Firstly, the drug product has to be deliberately designed to possess certain dimensions or to have special properties [43]. Secondly, it must possess at least one external dimension, or an internal or surface structure, in the nanoscale range (roughly 1 nm to 100 nm) [43]. Thirdly, the product should possess attributes or phenomena (physical, chemical, or biological effects, for example) that can be attributed to its nanoscale size [43]. Finally, even end products or materials with sizes beyond the conventional nanoscale range (up to 1000 nm) can be referred to as nanotechnology-based if the nanoscale-like properties are directly attributable to their engineered size [43].

From this analysis, we suggest a working definition of a nanodrug by combining the FDA [43] definition of nanotechnology products and the National Cancer Institute definition of a drug [44]. We thus define a nanodrug as: *A substance that involves the application of nanotechnology—as defined by the FDA—and is aimed at use in the prevention, diagnosis, treatment, or mitigation of the symptoms of a disease or abnormal physiological condition.*

We find it necessary to have a concise definition of a nanodrug to start with. Such a definition offers a shared paradigm within which the systematic and unbiased determination of the pharmaceutical products that have been approved so far, and which encompass nanotechnology can be conducted. It also allows for the categorization of approved nanodrugs by the type of nanomaterials incorporated in their formulations.

Bearing in mind the terminology of nanodrug as defined above, we systematically searched the literature to find clinically approved nanodrugs [4,5,6,9,10,26,28,45,46,47]. We found that 25 approved drugs meet the criteria laid down by our definition. Additionally, approximately 10 other nanodrugs have been approved on a national scale in single countries—for example, CosmoFer and Monofer, which have been approved in certain member states of the EU [4,5,6,9,10,26,28,45,46,47]. However, for the purpose of analysis, we considered only nanodrugs approved by either FDA or EMA. The latter are presented in Table 1.

Our analysis (Table 1) clearly demonstrates that lipid-based, polymer-based, and inorganic NPs are the most commonly used nanocarriers in the formulation of clinically approved nanodrugs (Figure 1). Their structural and functional flexibility makes them capable of delivering a wide range of therapeutic molecules, including small-molecule drugs, nucleic acids, and antibodies [4,5,6,9,10,26,28,45,46,47].

These nanocarriers (Figure 1) enhance conventional therapies by enabling more targeted and efficient delivery, thereby enhancing therapeutic efficacy and minimizing potential side effects [4,5,6,9,10,26,28,45,46,47]. Notably, they break biological barriers, improve the solubility of poorly water-soluble drugs, and prolong blood circulation time [4,5,6,9,10,26,28,45,46,47]. Thanks to these advantages, these nanocarriers are applied in many therapeutic areas with exceptional significance in cancer treatment and the management of chronic diseases [4,5,6,9,10,26,28,45,46,47]. Next, we briefly describe each category of nanocarriers.

### 3.2. Lipid-Based NPs

Lipid-based NPs are a prominent class of drug delivery systems composed primarily of lipids that self-assemble into nanoscale structures. They play an important role in drug delivery applications [4,9,28], as well as in gene therapy involving nucleic acids such as mRNA, siRNA, and DNA [4,5,6,9,10,26,28,45,46,47]. These NPs are typically spherical and consist of at least one phospholipid bilayer surrounding an aqueous core. Lipid-based NPs are widely utilized due to their many advantages in biological systems, including excellent biocompatibility and biodegradability, high drug-loading capacity, and a broad range of tunable physicochemical properties [65]. Moreover, they can reduce systemic toxicity while enhancing the efficacy and specificity of therapeutic agents [4,5,6,9,10,26,28,45,46,47]. As a result, lipid-based nanoparticles represent the most commonly used class among clinically approved nanomedicines (Table 1) [4,5,6,9,10,26,28,45,46,47].

However, these systems are not without limitations. They can trigger immune responses [66,67], and have been associated with increased rates of programmed cell death and potential degradation of nucleic acids such as DNA [68], which may be undesirable depending on the therapeutic context.

*Liposomes* are a well-established and extensively studied subclass of lipid-based NPs. They were first described in the 1960s [9,27,50] and remain among the most versatile drug delivery platforms [69]. Structurally, liposomes are unilamellar or multilamellar vesicles that mimic biological cell membranes. The phospholipid bilayer allows for the encapsulation of hydrophobic drugs, while the aqueous core is suitable for hydrophilic compounds [70]. Furthermore, the aqueous interior can be used to carry other therapeutic or diagnostic agents such as nucleic acids, proteins, or imaging compounds [71,72]. Several factors influence liposome stability, including particle size, shape, surface charge, lipid composition, and surface characteristics. Surface modifications, such as PEGylation, reduce opsonization and help prevent rapid clearance by the reticuloendothelial system, thereby extending circulation time and enhancing drug delivery efficiency [4,71,72]. Thanks to their versatility and established safety profiles, liposomes are widely used in clinical applications, including as carriers for chemotherapeutics, antibiotics, and gene therapy agents [4,71,72,73].

*Lipid nanoparticles (LNPs)* are a newer generation of lipid-based delivery systems, particularly suited for nucleic acid therapeutics [74]. Unlike liposomes, LNPs are typically non-lamellar and possess a micellar internal structure. Their formulation usually consists of four primary components: (1) a cationic or ionizable lipid for complexation with negatively charged nucleic acids, (2) phospholipids to maintain structural integrity, (3) cholesterol to improve particle stability and facilitate membrane fusion, and (4) PEGylated lipids to enhance circulation time, reduce aggregation, and control particle size [75]. LNPs are highly effective for nucleic acid delivery and can be engineered for targeted transfection [76,77]. Nevertheless, they face some limitations, including a generally lower loading capacity compared to liposomes and challenges in achieving optimal biodistribution. In particular, LNPs may accumulate in the liver and spleen, which could limit their therapeutic applications [10,74].

### 3.3. Polymer-Based NPs

Polymeric NPs are nanoscale drug delivery systems synthesized from natural or synthetic polymers [4,5,6,9,10,26,28,45,46,47]. Widely applied in medicine and pharmaceuticals, they are used for targeted drug delivery, cancer therapy, gene therapy, and diagnostics [4,5,6,9,10,26,28,45,46,47]. Their appeal lies in their biocompatibility, biodegradability, and formulation flexibility, which allows for diverse structural configurations and physicochemical properties [4,5,6,9,10,26,28,45,46,47]. This flexibility enables targeted delivery, controlled release, and surface modification, making polymeric NPs ideal for co-delivery applications [78] and precise control over drug loading and release kinetics [79]. Therapeutic agents can be incorporated via multiple strategies: encapsulation, matrix entrapment, chemical conjugation, or surface adsorption [4,5,6,9,10,26,28,45,46,47]. However, limitations include potential toxicity, nanoparticle aggregation, and the formation of toxic degradation by-products [4,5,6,9,10,26,28,45,46,47].

Several methods are available to tailor polymeric NP characteristics for specific applications. *Emulsification* involves dissolving a polymer and drug in a volatile organic solvent (e.g., dichloromethane), then emulsifying it into an aqueous phase [80,81]. Solvent evaporation leads to the formation of solid NPs, often used to encapsulate hydrophobic drugs. *Nanoprecipitation* entails dissolving the polymer and drug in a water-miscible solvent (e.g., acetone) and adding it to water under stirring, leading to spontaneous nanoparticle formation [82]. This method is efficient for small molecules and hydrophobic drugs. *Ionic gelation* is primarily used with natural polymers like chitosan [83,84]. The polymer is dissolved in water and mixed with a crosslinking agent such as tripolyphosphate, forming nanoparticles suitable for encapsulating biologically active compounds [83,84]. *Microfluidics* offers precise control over formulation by manipulating fluid streams in microchannels [85,86]. This technique enables high reproducibility and customization of nanoparticles.

Polymeric NPs are available in several structural forms, each suited to specific therapeutic goals, e.g., nanospheres, nanocapsules, dendrimers, micelles, polymersomes, nanohydrogels [4,5,6,9,10,26,28,45,46,47]. *Nanospheres* represent solid matrix systems in which drugs are uniformly dispersed, offering sustained or controlled release [4,5,6,9,10,26,28,45,46,47]. *Nanocapsules* feature a central core containing the active agent, surrounded by a polymeric shell [4,5,6,9,10,26,28,45,46,47]. These are especially effective for protecting sensitive drugs and enabling targeted release. *Dendrimers* are highly branched macromolecules with a central core and layered branches [87]. They offer multiple surface sites for attaching drugs, ligands, or imaging agents, and are widely used for delivering nucleic acids and small molecules. Dendrimers allow precise control over size, shape, molecular weight, and surface chemistry [88]. *Micelles* are self-assembled structures from amphiphilic block copolymers, consisting of a hydrophobic core and hydrophilic shell [89]. Ideal for encapsulating poorly water-soluble drugs, they improve bioavailability and efficacy [4,5,6,9,10,26,28,45,46,47]. *Polymersomes* are vesicle-like carriers composed of block copolymers, structurally similar to liposomes but with greater stability and loading capacity [90]. They are especially suited for dual drug delivery, as hydrophobic drugs can be integrated into the membrane and hydrophilic agents encapsulated in the core [91,92]. They also demonstrate efficient cytosolic delivery [93]. However, clinical translation remains limited due to production complexity and potential toxic by-products [94]. *Nanohydrogels* are three-dimensional hydrophilic polymer networks that swell in water [95]. They offer excellent biocompatibility and are well-suited for the controlled release of biomolecules, with promising applications in scaffold development and wound healing [96].

### 3.4. Inorganic NPs

Inorganic NPs are composed of materials such as metals (e.g., gold), metal oxides (e.g., iron), semiconductors (e.g., quantum dots), or silica [97]. Their main advantage lies in their tunable sizes, structures, and geometries, as well as their distinct optical, magnetic, electrical, and physical properties. These unique characteristics make inorganic NPs ideal for multimodal, stimuli-responsive, and targeted imaging applications [98]. Depending on their core composition, inorganic NPs serve different purposes. Metallic and silica NPs are commonly explored for drug delivery, thanks to their structural stability and high potential for surface functionalization [97]. Metal oxide NPs, particularly iron oxides, are used as magnetic resonance imaging (MRI) contrast agents due to their superparamagnetic properties, though many have been withdrawn from the market for safety or efficacy reasons [26,97].

To date, iron oxide nanoparticles remain the only FDA-approved inorganic NPs for clinical use (Table 1). Despite their potential, inorganic NPs face significant barriers to clinical translation, including low solubility, oxidative stress induction, and genotoxicity or cytotoxicity linked to heavy metal content [10,26,99]. Interestingly, in addition to iron oxide NPs, the EMA has recently approved hafnium oxide NPs for the treatment of squamous cell carcinoma (see Table 1) [64,100].

Generally, *metal nanoparticles* (e.g., gold, silver, platinum, and copper) interact strongly with light, making them valuable in diagnostics and imaging [4,5,6,9,10,26,28,45,46,47]. Their surfaces can be easily functionalized with biomolecules, enabling targeted delivery [4,5,6,9,10,26,28,45,46,47]. *Gold nanoparticles* are the most extensively studied, and can be synthesized in various morphologies, including nanospheres, nanorods, nanostars, nanoshells, and nanocages [101]. Their size and shape play a critical role in determining both their optical behavior and biological interactions [102]. Gold nanoparticles show good stability in biological environments, are generally non-toxicity and chemically inert [101]. Strong affinity for functional groups like thiols (-SH) and amines (-NH_2_), enabling conjugation with polyethylene glycol (PEG), proteins, or drugs [101]. However, despite their promise, no gold NP-based nanodrugs have received full regulatory approval, primarily due to unresolved issues regarding long-term biodistribution, clearance, and potential gene-related toxicity [103]. Nonetheless, several formulations are currently in clinical trials [104,105,106].

In contrast, *iron oxide NPs*-typically composed of magnetite (Fe_3_O_4_) or maghemite (Fe_2_O_3_)—are more broadly represented among FDA-approved nanomedicines [4,5,6,9,10,26,28,45,46,47]. Their approval largely stems from their use in treating iron deficiency in conditions such as anemia and chronic kidney disease (CKD) (Table 1), as their iron content is metabolizable by the human body. Iron oxide NPs are superparamagnetic, making them well-suited for MRI [107]. Additionally, they are being explored for drug delivery, thermal therapies (due to their ability to produce localized heating under an external magnetic field) and antimicrobial applications (although research is still ongoing, and evidence remains preliminary) [108,109].

## 4. General Cytotoxic Characteristics of Nanomaterials

The interactions between nanomaterials and biological systems are governed by a complex interplay of intrinsic physicochemical properties that are fundamental to the design and function of the nanomaterials.

Key among these is the chemical composition of the nanoparticle core, the presence of trace impurities, and the specific strategies used for surface functionalization (Figure 2). Surface characteristics (including surface charge, hydrophobicity, and the attachment of targeting ligands) further modulate the biological identity and reactivity of nanomaterials. Apart from the surface properties, size and shape are also critical determinants of the biological activity of nanoparticles and influence cellular uptake, biodistribution, and clearing mechanisms. Nanomaterials agglomerate in biological environments as well, greatly altering their effective size and surface area, and consequently their interactions with cells and tissues (Figure 2). Another key factor is biomolecular corona formation, an evolving layer of protein and other biomolecules that adsorbs onto the nanoparticle surface when exposed to biological fluid (Figure 2). The corona remodels the surface properties of the nanoparticle and is essential for mediating its detection and processing by the immune system and other cell components. Taken together, these characteristics determine the nano-bio interface and hence the pharmacokinetics, biodistribution, cellular interactions, and potential toxicological impact of nanomaterials on biological systems.

To comprehensively evaluate the toxic potential of clinically approved nanodrugs, it is crucial first to develop a clear idea of the general principles of nanomaterial-induced cytotoxicity. This means critically examining the physicochemical properties and biological processes that govern the cellular responses towards nanomaterials. In this regard, this section will present an overview of the most significant factors (such as particle size, shape, surface charge, composition, and functionalization) engaged in their cytotoxicity. In outlining these basic components, we wish to create the correct context for the interpretation of toxicity profiles evidenced in clinical application.

### 4.1. Cytotoxic Effects Determined by Nanomaterial Chemical Composition

The chemical composition of nanomaterials is a key determinant of their toxicity, as it governs critical properties such as solubility, redox activity, ionization behavior, and interactions with biological macromolecules [6]. These characteristics collectively shape their toxicological profile and help elucidate the mechanisms by which nanomaterials may induce adverse biological effects [6]. For instance, studies on zinc oxide nanoparticles (nano-ZnO) have demonstrated embryotoxic effects, primarily due to their dissolution into toxic Zn^2+^ ions [110]. Indeed, a recent comprehensive study evaluated the toxicity of 29 rare earth oxide and transition metal oxide nanoparticles in liver cells [111]. The results showed that pro-oxidative transition metal oxide NPs induced apoptosis in hepatocytes, while rare earth oxide NPs triggered pyroptosis in Kupffer cells and other macrophages [111]. These distinct cell death pathways are likely driven by differences in solubility, redox behavior, and catalytic activity among the nanoparticles [111,112]. However, a common feature underlying the toxic response was the excessive generation of reactive oxygen species (ROS) [111]. Depending on their capacity to induce ROS, nanoparticles triggered distinct cytotoxic pathways [111].

While certain nanomaterials (such as copper oxide, silver NPs, and quantum dots) pose toxicity risks due to the release of harmful metal ions [113], others, like gold NPs, demonstrate significantly greater biocompatibility [101]. For instance, gold NPs have been shown to be non-toxic to RAW264.7 macrophages and can even reduce ROS levels, suggesting potential anti-inflammatory properties [114]. In contrast, soluble metallic nanoparticles tend to increase ROS generation and cytotoxicity due to their ion-releasing behavior [111]. Non-metallic nanomaterials, such as mesoporous silica nanoparticles, generally exhibit lower toxicity [6]. This is largely attributed to the presence of surface silanol groups that modulate interactions with biological molecules [115]. Similarly, liposomes—due to their excellent biocompatibility—have been successfully employed in clinical settings as drug carriers, enhancing therapeutic efficacy while minimizing side effects [4,5,6,9,10,26,28,45,46,47].

Furthermore, impurities also play a critical role in determining nanomaterial toxicity. For instance, residual metal catalysts in carbon nanotubes (CNTs) can enhance ROS production, contributing to adverse biological effects [6,116]. Similarly, gold nanoparticles synthesized with cetyltrimethylammonium bromide (CTAB) can be toxic if residual CTAB is not adequately removed [117]. Organic contaminants, particularly endotoxins like lipopolysaccharides (LPS), are another source of concern [6]. LPS are resistant to standard sterilization and can activate Toll-like receptor 4 (TLR4) on immune cells, triggering the release of pro-inflammatory cytokines [6]. Therefore, accurate nanotoxicity assessment must include rigorous screening and removal of LPS to avoid misinterpretation of inflammatory potential [118,119].

### 4.2. Cytotoxic Effects Determined by Nanomaterial Size

Particle size is another determinant in the interaction between NPs and biological systems [120,121,122]. It influences not only cellular uptake but also distribution, diffusion, and biological reactivity [123]. Smaller NPs have a higher surface area-to-volume ratio, enhancing their catalytic activity and interaction with biological environments [124,125]. For instance, gold NPs of 10-16 nm can penetrate the cytoplasm, while particles <6 nm may even reach the cell nucleus, increasing potential toxicity [126].

Although NPs around 50 nm are considered optimal for intracellular uptake, both smaller (15–30 nm) and larger (70–240 nm) particles tend to show reduced internalization [127]. A consensus has now been reached regarding the primary mechanisms by which NPs enter cells [128]. There are five major pathways: phagocytosis, macropinocytosis, clathrin-mediated endocytosis (CME), fast endophilin-mediated endocytosis (FEME), and clathrin-independent carrier/glycosylphosphatidylinositol-anchored protein-enriched early endocytic compartment (CLIC/GEEC) endocytosis [128]. CME, FEME, CLIC/GEEC pathways typically internalize NPs in the 60–100 nm range, while macropinocytosis and phagocytosis accommodates larger particles >250 nm via large vesicles (0.5–1.5 μm) [128,129,130]. In addition to size, surface characteristics such as hydrophobicity, ligand presence, and the cell type also influence uptake [128,129,130]. For example, Herceptin-conjugated gold NPs (40–50 nm) demonstrated enhanced cellular internalization compared to both smaller (2–10 nm) and larger (80–100 nm) particles, underscoring the role of ligand density, binding affinity, and membrane wrapping in cellular uptake [131].

The ideal NP size for biomedical applications generally ranges from 10 to 100 nm, with ~50 nm being frequently used for drug delivery due to efficient uptake and favorable pharmacokinetics [132,133]. Size also plays a pivotal role in systemic and organ-specific responses [132,133]. For instance, particles between 10 and 50 nm can be efficiently cleared by the kidneys, whereas larger particles are more likely to accumulate in organs like the liver, spleen, or lungs [26,132,133,134].

Toxicological studies confirm that smaller NPs, such as 20 nm TiO_2_, induce stronger inflammatory responses in the lungs compared to larger particles (250 nm), largely due to their greater surface area and reactivity [135,136]. Furthermore, particles <100 nm are more prone to tissue accumulation, leading to potential oxidative stress, inflammation, and organ toxicity [6,137]. Recent findings also suggest that prolonged exposure to small NPs may cause histopathological changes, particularly in the liver [6,10,26]. Notably, silica NPs exhibited higher hepatotoxicity compared to silver or zinc oxide NP composites [138,139].

### 4.3. Cytotoxic Effects Determined by Nanomaterial Shape

NPs exhibit a variety of shapes—such as nanospheres, nanotubes, nanosheets, nanowires, and nanocubes—which significantly influence their toxicokinetics and cytotoxicity [6,140]. Shape affects cellular uptake, cytotoxicity, biodegradation, and organ-specific toxic effects [6]. Computational simulations suggest that nanospheres are internalized most efficiently via endocytosis, followed by nanocubes, nanotubes, and nanodiscs, likely due to the lower membrane deformation energy required by highly symmetrical structures [121,141]. Experimental studies on gold NPs support this, showing that nanospheres result in the lowest cell viability, with prismatic gold NPs slightly less toxic [6,123]. Cubic and rod-shaped gold NPs demonstrated similar bioavailability across a size range of 10–100 nm [123]. Despite reduced internalization, some gold NPs induced notable cytotoxicity due to mechanical damage from sharp edges [6,142].

In mesoporous silica NPs, adjusting the aspect ratio of rod-shaped particles has been shown to decrease hepatic accumulation and enhance renal clearance, regardless of administration route [6]. While oral administration of mesoporous silica NPs typically results in minimal organ toxicity, shape-dependent nephrotoxicity remains a concern and may cause significant kidney injury [143,144].

It is worth noting here that particle shape also influences the exposure of crystallographic planes, biocorona formation, and catalytic activity—all factors contributing to the overall toxicity profile [6,145].

### 4.4. Cytotoxic Effects Determined by Nanomaterial Surface Properties

Nanomaterials interact with biological systems primarily through their surface, making surface characteristics an important determinant of their behavior and toxicity [145]. Critical surface properties include charge, hydrophobicity, and the presence of specific functional groups or atoms [6,145]. In biomedical applications, NPs are frequently modified post-synthesis to enhance colloidal stability, biocompatibility, safety, and pharmacokinetics [6,145]. These modifications significantly influence the nanomaterials’ interfacial behavior and toxicological profile [145,146]. For instance, rendering NP surfaces hydrophilic and electro-neutral can improve their ability to traverse mucus barriers by reducing interactions with negatively charged, hydrophobic mucin fibers [6,145,147]. In contrast, hydrophobic and positively charged NPs more effectively penetrate epithelial barriers [6,145,147]. Therefore, surface properties must be tailored to match the target biological environment for optimal absorption and delivery [6,145,147,148].

Surface features also influence biodistribution and excretion by affecting protein interactions, biocorona formation, and colloidal stability of NPs [6,149]. For example, PEG-coated gold NPs show enhanced stability in the bloodstream and more uniform tissue distribution [4,6,101,114,149]. Conversely, polyethyleneimine (PEI)-coated gold NPs tend to aggregate and are rapidly cleared by the mononuclear phagocyte system (MPS), leading to accumulation in the liver and spleen and reduced systemic availability [4,6,101,114,149].

Surface charge also affects cellular uptake and intracellular trafficking of NPs [6,123]. Positively charged NPs are internalized more readily due to strong electrostatic interactions with the negatively charged cell membrane [150,151]. Surface ligands further enhance targeting by directing NPs to specific cell types or subcellular compartments [152,153]. Additionally, surface coatings can modulate oxidative stress executed by NPs. For example, silver NPs coated with PEG or bovine serum albumin (BSA) generate fewer ROS and exhibit reduced cytotoxicity then noncoated NPs [154,155].

### 4.5. Cytotoxicity Driven by Aggregation/Agglomeration

Nanomaterial aggregation and agglomeration is another factor influencing cytotoxicity, with emphasis on effects on biodistribution and cellular uptake [6]. Generally, agglomerates are defined as loosely bound particles and aggregates—as chemically or tightly bound particles [6]. In fact, the toxicological behavior of aggregates and agglomerates is not yet fully understood and warrants further investigation [156,157,158].

Agglomerated nanomaterials may enter cells via different internalization pathways compared to individual nanoparticles, leading to altered interactions with cellular receptors and membrane proteins [159]. Both aggregates and agglomerates have demonstrated notable cytotoxic potential in biological systems [158,159,160]. The biological effects of aggregated or agglomerated nanomaterials are closely tied to their physicochemical characteristics, which shape their interactions with cells, proteins, and tissues, ultimately influencing their toxicological outcomes [110,158,159,160,161].

### 4.6. Impact of Biocorona Formation on Cytotoxicity

The protein corona, or biocorona, forms rapidly when nanoparticles come into contact with biological fluids, resulting in the adsorption of various biomolecules onto their surface [162,163,164,165]. This dynamic layer redefines the NP’s biological identity and significantly alters its interactions with cellular systems [162,163,164,165]. The biocorona critically influences the biokinetics and biological fate of nanoparticles, including their biodistribution and clearance [162,163,164,165,166,167]. For example, glucose-coated iron oxide NPs preferentially accumulate in the liver, while PEG-coated counterparts demonstrate broader tissue distribution [6,162,163,164,165,166]. These differences stem from variations in biocorona composition, affecting degradation pathways and macrophage-mediated uptake [6,162,163,164,165,166].

At the cellular level, the biocorona reduces nonspecific interactions with membranes, preserving membrane integrity and modifying internalization routes [6,162,163,164,165,166]. Unlike bare nanoparticles, coronated NPs are less likely to disrupt the lipid bilayer directly, which mitigates membrane damage and can improve biocompatibility [168,169]. Moreover, the biocorona plays a pivotal role in modulating cytotoxicity by altering NP bioavailability, immune recognition, and intracellular interactions [170,171,172]. It can either mask or expose reactive surface sites, influencing the extent of oxidative stress, inflammation, or augment cell death execution triggered by NPs [6,162,163,164,165,166]. In summary, the formation of the protein corona is one of the main determinants of NPs behavior in biological systems, governing not only cellular uptake mechanisms but also the safety and efficacy of nanomaterials in biomedical applications.

### 4.7. Effects of Degradation and Metabolization Products on Cytotoxicity

Other players that largely influence cytotoxicity of nanomaterials are the degradation of nanomaterials and their metabolic byproducts, along with the long-term implications of their biotransformation in biological systems [6]. Once internalized by cells, nanomaterials are subjected to harsh biochemical environments, notably enzymes in hepatic cells and within the acidic, oxidative, and ion-rich environment of lysosomes—particularly in cells of the MPS [10,26,134]. These compartments are rich in enzymes, acids, oxidative agents, and metal ions, all of which facilitate the breakdown or transformation of nanomaterials [10,26,134,172,173,174].

The extent, rate, and pathway of nanomaterial degradation significantly influence their toxicological profile [6,10,26,134,172,173,174]. These processes alter the materials’ physicochemical characteristics (including size, surface chemistry, and solubility) which in turn affect their biodistribution, cellular uptake, and interaction with biomolecules [6,10,26,116,134,172,173,174]. Importantly, degradation products themselves may exhibit distinct or enhanced toxic effects, depending on their chemical composition, reactivity, and accumulation in target tissues [6,10,26,116,134,172,173,174]. For instance, mesoporous silica NPs with a lower aspect ratio degrade more rapidly in simulated body fluids, leading to increased systemic absorption, preferential hepatic accumulation, and severe renal toxicity following oral exposure [143].

One of the most extensively studied cases of degradation-driven toxicity involves silver NPs [6]. Their primary toxic mechanism is linked to ion release through lysosomal dissolution [6,139,154,175]. In the acidic environment of lysosomes, silver NPs readily dissolve into Ag^+^ ions [6,139,154,175]. This dissolution disrupts lysosomal membrane integrity, resulting in leakage of both silver ions and lysosomal enzymes into the cytoplasm [6,139,154,175,176]. Once in the cytosol, Ag^+^ ions contribute to mitochondrial dysfunction, oxidative stress, and apoptosis induction [6,139,154,175,176,177]. Furthermore, these ions can react with organic acids to form Ag-O-complexes, which may subsequently convert into Ag-S-species through interactions with thiol-containing proteins, such as those containing cysteine residues [6,139,154,175,176,177]. These chemical transformations exacerbate mitochondria-mediated apoptosis and influence the intracellular retention or excretion of silver [6,139,154,175,176,177,178].

Understanding nanomaterial degradation is essential for predicting long-term biological effects, especially for materials designed for chronic exposure or therapeutic use [6]. Accumulation of toxic byproducts, organ-specific degradation kinetics, and incomplete clearance can all contribute to persistent cytotoxicity or delayed onset of adverse outcomes [6,10,26]. Therefore, both the parent nanomaterial and its degradation/metabolic products must be considered when assessing safety and biocompatibility [6].

## 5. Defining Molecular Determinants of Nanodrugs’ Cytotoxic Effects

Having defined the term *nanodrug*, outlined the major categories of clinically approved nanodrugs, and examined the general cytotoxic properties of nanomaterials, we proceeded to conduct a systematic literature search, screening, and data extraction focused on nanodrugs (using the definition provided in Section 3.1) and their associated cytotoxicity. To assist in this process, we employed Elicit AI Pro, an AI-powered research tool, to identify and extract relevant molecular determinants underlying nanodrug-induced cytotoxic effects. A comprehensive description of how Elicit AI Pro was utilized for data extraction and analysis is provided in Section 2.

### 5.1. Key Molecular Pathways Determining Cytotoxicity of Nanodrugs

Using Elicit AI Pro, we analyzed a broad range of studies that consistently highlight several key molecular mechanisms driving nanoparticle-induced cytotoxicity. These mechanisms are often interconnected and may trigger cascading cellular effects, amplifying toxic outcomes. A summary of the principal molecular pathways identified as contributors to nanoparticle-induced cytotoxicity is presented in Table 2.

It is worth noting that our analysis revealed that nanodrug platforms exhibit formulation-dependent cytotoxicity profiles, with active targeting strategies generally outperforming passive approaches in terms of tumor selectivity and therapeutic efficacy [6,10,35,110,138,142,179,184,188]. Active targeting uses receptor-mediated endocytosis, enabling enhanced uptake of nanocarriers by tumor cells and facilitating drug delivery at lower concentrations [6,10,35,110,138,142,179,184,188]. Clinical studies of approved nanodrugs have revealed notable differences in cytotoxicity and molecular interaction profiles across formulations [6,10,35,110,138,142,179,184,188]. Nanocarriers functionalized with active targeting ligands—such as aptamers, folate, transferrin, and prostate-specific membrane antigen—have demonstrated increased cellular internalization and induced apoptosis in tumor cells at significantly lower drug doses compared to their non-targeted counterparts [189]. For example, apoferritin nanocages achieved IC_50_ values ranging from 14 to 225 nM, compared to 14 to 68 nM for the free drug, suggesting a concentration-dependent but improved delivery efficiency [190]. Carbon nitride dot-based platforms induced apoptosis at 50 nM, markedly outperforming free doxorubicin, which required concentrations up to 1000 nM [190]. Liposomal formulations and polymer-drug conjugates demonstrated IC_50_ values between 20 and 2000 nM, often exhibiting reversible or attenuated cytotoxicity, improving safety margins [191]. Squalenoylated lipid nanoparticles allowed for a fivefold increase in the maximum tolerated dose, significantly reducing cardiotoxicity commonly associated with anthracycline-based therapies [192].

The mechanistic basis for these enhanced outcomes centers on specific molecular interactions between nanoparticle surfaces and tumor cell receptors [6,10,35,110,138,142,179,184,188]. Key receptors implicated in targeted delivery include heat shock protein 70 (HSP70), transferrin receptor 1 (TfR1), folate receptors, and integrins [6,10,35,110,138,142,179,184,188]. Targeting these molecules facilitates preferential tumor accumulation, minimizes systemic exposure, and reduces off-target toxicity [6,10,35,110,138,142,179,184,188]. In contrast, passively targeted nanodrugs rely on the enhanced permeability and retention (EPR) effect to accumulate in tumor tissue [193,194,195]. While this mechanism can be effective, it lacks the precision of receptor-mediated targeting and often results in broader biodistribution and higher systemic toxicity [193,194,195]. It is worth noting here that studies challenged the efficacy of EPR effect, identifying a high degree of heterogeneity and limited experimental data from patients on the effectiveness of this mechanism [193,194,195]. In fact, tumor microenvironment, e.g., vascular density, extracellular matrix composition, and interstitial pressure, are main factors in determining the extent of nanoparticle extravasation [193,194,195,196,197]. Imaging assessments and tailored nanocarrier design emerge as critical considerations to address the inherent heterogeneity of the EPR effect as reported in both preclinical and clinical settings [193,194,195]. Finally, therapeutically optimal cytotoxicity relies to a great extent on the targeting strategy and is largely characterized by the mechanism of action of the API. Typically, the data indicate that nanodrug preparations featuring well-established active targeting strategies are generally more superior regarding their selectivity toward tumors, cytotoxicity profiles, and therapeutic indexes than those relying entirely on passive targeting mechanisms.

While nanodrugs offer enhanced efficacy and targeted delivery, their cytotoxic potential can also contribute to adverse drug reactions (ADRs). Assessing and understanding nanodrug-induced toxicity is therefore essential for ensuring patient safety and guiding the development of safer nanopharmaceuticals. Studying cytotoxicity at the cellular and molecular levels enables researchers to pinpoint the specific pathways and mechanisms by which nanodrugs exert toxic effects. Such mechanistic insights are critical for predicting, mitigating, and ultimately preventing ADRs during both preclinical development and clinical application. To address this, we conducted a systematic analysis of current literature, using defined criteria for nanodrugs (as outlined in Section 3.1), with the aim of identifying shared molecular features and common signaling pathways linked to ADR-related cytotoxicity. This effort focuses on elucidating the cellular mechanisms most frequently implicated in the toxic responses to nanodrug formulations.

Our findings, summarized in Table 2, reveal that three major molecular mechanisms consistently underlie nanoparticle-induced cytotoxicity: oxidative stress pathways (characterized by excessive generation of ROS, leading to mitochondrial dysfunction, lipid peroxidation, and DNA damage), inflammatory signaling cascades (involving activation of pro-inflammatory cytokines and key pathways such as NF-κB and mitogen-activated protein kinases-MAPK, which contribute to chronic inflammation and tissue damage) and lysosomal dysfunction (resulting from nanoparticle accumulation or degradation within lysosomes, causing membrane destabilization, leakage of enzymes, and disruption of cellular homeostasis) (see details in Table 2).

*Oxidative stress* consistently emerges as a predominant mechanism of nanotoxicity across numerous studies (Figure 3) [6,10,35,120,122,136,138,176,179,184]. Alongside inflammation and DNA damage, oxidative stress has been identified as a key contributor to the cytotoxic effects of nanodrugs (Figure 3) [6,10,35,120,122,136,138,176,179,184]. A central driver of oxidative stress is the excessive generation and/or accumulation of ROS (highly reactive molecules that can damage essential cellular components) [6,10,35,120,122,136,138,176,179,184]. The specific mechanisms by which nanoparticles induce ROS production vary depending on their physicochemical properties, including size, shape, surface charge, and material composition [6,10,35,120,122,136,138,176,179,184]. Some nanoparticles directly produce ROS through surface redox reactions or photoactivation, while others indirectly increase ROS levels by disrupting cellular antioxidant defenses (Figure 3) [179,198].

For example, platinum NPs have been shown to mimic the activity of NADPH oxidase, an enzyme complex involved in ROS generation, potentially amplifying oxidative stress in exposed cells [198]. However, multiple studies coherently report that rising levels of ROS compromise redox balance [6,10,35,120,122,136,138,176,179,184]. Elevated ROS levels coincide with glutathione depletion, loss of mitochondrial membrane potential, lipid peroxidation, and damage to DNA and proteins (Figure 3) [6,10,35,120,122,136,138,176,179,184]. Thus, current evidence indicates that nanodrug-induced oxidative stress mediates cytotoxicity primarily via ROS generation and impaired antioxidant defenses (Figure 3) [6,10,35,120,122,136,138,176,179,184].

*Inflammation* represents another major mechanism of nanoparticle-induced toxicity and is frequently observed alongside oxidative stress and DNA damage (Figure 4) [6,10,35,120,122,136,138,176,179,184]. Numerous studies have demonstrated that NPs can activate a wide array of inflammatory signaling pathways, most notably NF-κB, MAPK, and phosphatidylinositol 3-kinase/Akt (PI3K/Akt) pathways [179,199]. These activations result in the upregulation and release of pro-inflammatory cytokines such as TNF-α, IL-6, and IL-8, contributing to both local and systemic inflammatory responses (Figure 4) [179]. Exposure to nanoparticles has been shown to modulate gene and protein expression in inflammatory cascades, including cyclooxygenase-2 (COX-2) and downstream MAPK/PI3K/Akt pathways [199]. This dysregulation contributes to an environment of chronic inflammation, which can drive further cytotoxicity and impair cellular homeostasis [6,10,35,120,122,136,138,176,179,184]. A particularly well-characterized mechanism involves the activation of the NLR family pyrin domain containing 3 (NLRP3) inflammasome, especially in immune and epithelial cells [200,201,202,203,204]. For instance, amino-modified polystyrene NPs have been shown to destabilize lysosomes and induce ROS production, both of which contribute to NLRP3 inflammasome activation [200,201,202,203,204]. This leads to the release of interleukin-1β (IL-1β) and initiates pyroptotic or necrotic cell death pathways (Figure 4) [200,201,202,203,204]. Similarly, zinc oxide NPs have been reported to stimulate NF-κB and MAPK signaling in monocytes, leading to the production of TNF-α and IL-1β, enhanced oxidative stress, and DNA damage [205]. Additional studies highlight shifts in cytokine profiles—including increased levels of nitric oxide (NO), IL-6, and inducible nitric oxide synthase (iNOS)—and changes in macrophage polarization, which further link inflammation to nanoparticle-mediated cytotoxicity [206].

Importantly, a growing body of research suggests that these inflammatory effects can be modulated by therapeutic interventions [200,201,202,203,204]. The use of ROS scavengers, nonsteroidal anti-inflammatory drugs (NSAIDs), small-molecule pathway inhibitors, and optimized nanoparticle formulations has shown promise in attenuating NP-induced inflammation and mitigating associated cytotoxicity [200,201,202,203,204,207]. Therefore, the relationship between oxidative stress and inflammation is particularly noteworthy, as ROS generation frequently acts as a trigger for inflammatory cascades. This creates a positive feedback loop in which inflammation enhances ROS production, and vice versa, amplifying cellular stress, injury, and death [200,201,202,203,204].

Next, *lysosomal dysfunction* has emerged as a critical intracellular mechanism underlying nanodrug-induced cytotoxicity (Figure 5). Numerous studies have identified lysosomal damage, membrane permeabilization, and impaired enzymatic activity as common cellular responses to nanoparticle exposure (Figure 5) [6,10,35,120,122,136,138,176,179,184]. Lysosomes, essential for cellular homeostasis, act as key sites for the degradation and recycling of biomolecules [208]. However, due to their acidic environment and membrane-bound architecture, they are also particularly vulnerable to disruption by internalized nanomaterials (Figure 5) [6,10,35,120,122,136,138,176,179,184]. The mechanisms by which nanoparticles induce lysosomal dysfunction are diverse and highly dependent on particle composition, surface charge, and degradation behavior [6,10,35,120,122,136,138,176,179,184]. Certain nanoparticles—particularly those with cationic surface charges or acid-sensitive moieties—tend to accumulate within lysosomes [10,180]. This accumulation can lead to lysosomal membrane permeabilization (LMP), leakage of hydrolytic enzymes such as cathepsins into the cytoplasm, and subsequent activation of cell death pathways, including apoptosis and necrosis (Figure 5) [10,180,209]. Other nanoparticles may disrupt lysosomal pH regulation, impair autophagic flux, or interfere with exocytosis, thereby disturbing the cell’s waste management and stress response systems (Figure 5) [181].

Studies have shown that cationic liposomes and pH-responsive nanogels can induce LMP, triggering cathepsin release, mitochondrial membrane depolarization, and excessive ROS generation [6,10,35,120,122,136,138,176,179,184,210]. For example, zinc oxide NPs have been reported to impair autophagy and reduce mitochondrial membrane potential—a process mitigated by antioxidant treatment such as N-acetylcysteine, implicating oxidative stress as a downstream consequence of lysosomal damage [211]. Similarly, iron oxide nanoparticles have been shown to promote ROS formation via Fenton-type reactions, leading to mitochondrial dysfunction, DNA damage, and enhanced cytotoxic or anticancer effects [26,212]. Notably, these events often follow or coincide with lysosomal disruption, reinforcing the idea that lysosomal dysfunction serves as both a trigger and amplifier of oxidative stress [10,26]. Lysosomal impairment triggered by NPs is typically characterized by membrane destabilization or rupture, inhibition of autophagic flux, disruption of lysosomal enzyme activity, altered exocytosis or intracellular trafficking (Figure 5) [6,10,35,120,122,136,138,176,179,184,210]. These events initiate a cascade of intracellular disturbances that ultimately result in cell death, as confirmed by markers of apoptosis, necrosis, and inflammatory responses [6,10,35,120,122,136,138,176,179,184,210]. Quantitative analyses, including IC_50_ values and biomarker profiling, further support the cytotoxic impact of lysosomal damage in various cell types [6,10,35,120,122,136,138,176,179,184,210].

Collectively, the evidence suggests that lysosomal dysfunction is a pivotal contributor to nanodrug cytotoxicity, closely interconnected with oxidative stress and mitochondrial impairment. This crosstalk between organelle systems exacerbates cellular damage and underscores the importance of lysosomal stability in the safe design of nanodrug platforms.

It is important to stress the need for summarizing representative research on cytotoxicity with close attention to material-specific trends, NP types, and their correlations with cytotoxic effects. While Table 2 outlines major molecular pathways in nanoparticle-induced cytotoxicity, it is a generalized picture lacking stratification by NP type. For this reason, we present Table 3 and Table 4, which summarizes representative research with particular attention to stratification by nanoparticle type.

### 5.2. Predictive Frameworks Determining Cytotoxicity of Nanodrugs

While oxidative stress, inflammation, and lysosomal destabilization are distinct molecular signatures of nanoparticle toxicity, previous reviews have primarily approached these mechanisms in general contexts or relative to preclinical material rather than clinically relevant nanodrug formulations [10,35,116,120,136,145,176,179,184]. The aim of this work is to move beyond repetition through the identification of formulation-dependent molecular mechanisms and uncovering under-valued mechanistic properties related to active versus passive targeting, carrier type, and intracellular trafficking behavior. In order to achieve this, we conducted a systematic re-review of the literature that solely focused on nanodrugs (defined as in Section 3.1), supplemented by Elicit platform-facilitated pathway extraction with AI support. This allowed us to compare which different nanodrug classes selectively modulate cytotoxic pathways, quantify biomarkers, and evaluate underexplored mechanisms like autophagy inhibition and inflammasome activation. Our aim was to develop an even finer-grained, comparative model of predicting and averting ADRs at the molecular level.

Having identified the key molecular drivers of cytotoxicity induced by nanodrugs (e.g., oxidative stress, damage and barrier dysfunction of the membrane, lysosomal impairment, activation of pro-inflammatory cascades, and mitochondrial injury) we applied the Elicit platform for data extraction from foundational literature (see Appendix A) and based on summarized data generated a qualitative comparative matrix. We constructed a qualitative comparative matrix (Table 4) that associates specific nanodrug classes (e.g., liposomes, polymer-drug conjugates, inorganic nanoparticles) with their most prevalent cytotoxic mechanisms (e.g., ROS generation, lysosomal membrane disruption, cytokine release). This enables comparison across formulation-type toxicological profiles and uncovers the relative importance of each molecular stimulus toward total cytotoxicity. We grouped each cytotoxic mechanism into three categories based on its reported association strength: strong (consistently reported or well-known as a first-order mechanism), moderate (consistently reported), and weak (occasionally reported).

This was conducted according to how frequently and prominently each mechanism appeared in the literature. The association strength was qualitatively assessed in the studies referenced. As many publications did not report specific IC_50_ values or quantitative ROS measurements, the resulting matrix is presented in a qualitative format.

It needs to be noted that due to the structural and compositional diversity of nanomaterials (i.e., metal, polymer, and lipid-based platforms) their interactions with biological systems, and the resulting toxic effects, are highly context-dependent [6,10,35,110,138,142,179,184,188]. They may vary with the route of administration, biological environment, and nanomaterial architecture [6,10,35,110,138,142,179,184,188,236]. As current evidence indicates, there could be no universally applicable nanodrug platform, given that cytotoxic mechanisms are largely determined by the kind of nanomaterial being used [10,21]. However, it is still possible to generalize existing cytotoxicity data by stratifying it into classes of nanomaterials, whereby common mechanistic trends are established for comparable platforms and even finding cross-platforms similarities.

Summarizing evidence in Section 4 and Section 5, it is obvious that cytotoxicity of nanomaterials is strongly influenced by numerous physicochemical parameters like particle size, charge on the surface, composition of the material, coating on the surface, and shape [6,10,35,110,138,142,179,184,188]. Below, we briefly outline each of these parameters’ contribution. There is a rapidly emerging consensus in the literature that the smaller NPs will be more cytotoxic compared to larger NPs [26,132,133,134]. For example, 1.4 nm gold NPs have been found to be toxic, while nanoparticles with diameters of 15 nm or larger are typically found to have negligible or no toxicity [237,238,239]. Such a size-dependent effect is largely due to increased surface area, cellular uptake, and increased interactions with intracellular components [6,10,35,110,138,142,179,184,188]. Positively charged NPs are found to show increased cytotoxicity due to increased electrostatic interactions with the negatively charged cell membranes [182]. These interactions can cause enhanced membrane disruption and facilitate cellular uptake, with the potential to produce down-stream toxicities [182]. Surface modifications such as PEGylation can prevent nonspecific interactions and cytotoxicity [240]. However, they also have the potential to introduce new problematics such as immunogenic reactions or alterations in biodistribution [240]. The presence and integrity of a protein corona, a biomolecular layer that develops on the NP surface when it comes into contact with biological fluids, also make significant impacts on cellular interactions, recognition, and immunity by the immune system [240]. Toxicity can be material-specific. For instance, cadmium-based quantum dots (e.g., CdTe) are more toxic than quantum dots that are protected by multiple layers (e.g., CdSe/ZnS) [187]. Metal oxide nanoparticles have been shown to produce ROS, which play a role in oxidative stress and cellular damage [241,242]. Carbon nanomaterials, such as carbon nanotubes, particularly needle-like structures, have been said to exhibit unusual and in most instances increased toxicity due to their shape and stiffness [110,116,185]. Hydrophobic surfaces also contribute to cell membrane disruption and hence become more cytotoxic [182]. The disruption may enable unregulated entry of nanoparticles into cells and organelles [182]. Additionally, size plays a crucial role in immune recognition and cellular uptake [6,10,35,110,138,142,179,184,188]. Furthermore, spherical NPs are more readily internalized by macrophages [186]. High-aspect-ratio needle-like carbon nanotubes, however, have been associated with toxicity, which could be due to frustrated phagocytosis and persistent cellular stress [243,244].

Based on the reviewed literature and our systematic analysis, we have summarized the key cytotoxicity mechanisms associated with different classes of nanodrugs (Figure 6). Each class of nanomaterial has characteristic biological interactions and cellular effects. Metal NPs are primarily recognized as potent generators of ROS, inducing oxidative stress, lipid peroxidation, DNA damage, and leading to apoptosis or necrosis (Figure 6). Quantum dots, by release of the heavy metal core components, show high mitochondrial toxicity by disrupting membrane potential, inhibiting ATP synthesis, and initiating intrinsic apoptotic pathways (Figure 6). Carbon nanotubes and lipid NPs will most likely cause robust inflammatory reactions, partly due to their geometry, surface characteristics, and immune recognition, often in combination with cytokine release and macrophage activation. Polymeric NPs tend to accumulate in lysosomes and cause lysosomal destabilization, which may lead to leakage of the enzyme into the cytoplasm and initiation of cell death pathways (Figure 6).

Based on our systematic analysis (supported by the data presented in Table 4 and the cytotoxicity mechanisms illustrated in Figure 6) we propose a semi-quantitative model for evaluating the cytotoxic risk of nanomaterials (Figure 7). This model, titled the *Toxicity Index Scoring System*, integrates representative physicochemical properties, biological response indicators, and surface modification parameters to give a composite risk score of 0 to 21 points. Depending on the overall score, nanomaterials receive low, moderate, or high cytotoxicity risk categories.

The compositional and structural characteristics of nanoparticles (particularly particle size, shape, surface charge, and material composition) are important determinants of their biological behavior. These parameters dictate the interactions of nanoparticles with cells, tissues, and organelles, and thus establish their toxicological profiles. In addition to these intrinsic properties, we incorporate biomarker-based parameters reflecting cellular responses commonly encountered in both in vitro and in vivo toxicological studies. These include the levels of oxidative stress, inflammatory cytokine release, and damage to organelles or membranes—each a functional indicator of nanoparticle-induced cytotoxicity. Finally, the modifying factors such as surface PEGylation, biodegradable coatings, or protein corona formation are also considered, as these may decrease or increase cytotoxic potential. Taken together, this structured *Toxicity Index Scoring System* constitutes a semi-quantitative, evidence-based framework for the direct comparison of nanoparticle cytotoxicity across different material types. It is designed to support preclinical safety screening, safer-by-design nanomaterials, and inform risk assessment strategies in the fields of nanomedicine, toxicology, and environmental health sciences.

## 6. Outlook and Perspectives

Our analysis reveals that several classes of nanodrugs—including liposomes, albumin-based nanospheres, and iron oxide nanoparticles—have already received FDA approval and are in clinical use, while others are progressing through various stages of clinical translation [4,5,6,9,10,26,28,45,46,47]. The choice of nanodrug platform and targeting strategy should be carefully tailored to tumor type, receptor expression patterns, and the broader clinical context to optimize therapeutic efficacy and safety [4,5,6,9,10,26,28,45,46,47].

We observed notable differences among nanodrugs in terms of cytotoxicity and molecular interactions. These variations arise from the complex interplay of multiple factors, including nanoparticle size, shape, surface charge, functionalization, and composition [6,10,26,116,134,172,173,174]. For instance, cadmium-containing quantum dots consistently demonstrate higher toxicity compared to other nanomaterials [187]. Additionally, many studies report a dose-dependent increase in toxicity, with higher concentrations of nanoparticles inducing more severe biological effects [199]. Despite the diversity of nanomaterials and their applications, our analysis identified three predominant molecular mechanisms that underlie nanoparticle-induced cytotoxicity: oxidative stress pathways, inflammatory signaling cascades, lysosomal dysfunction. These shared mechanisms are summarized in Table 2. Understanding these pathways provides a foundation for the rational design of risk mitigation strategies. We outlined tentatively proposed risk mitigation strategies in Table 5 that are tailored to nanoparticle type and cytotoxic mechanisms.

A recurring concern across different nanodrug categories is the generation of cytotoxic ROS, which is closely linked to oxidative stress and often acts as a central mediator of cytotoxicity [6,10,35,120,122,136,138,176,179,184]. ROS generation is frequently observed across a wide range of nanomaterials and often intersects with inflammatory and lysosomal pathways, forming feedback loops that amplify cellular damage [6,10,35,120,122,136,138,176,179,184]. This interplay is visualized in Figure 3, Figure 4, Figure 5 and Figure 6 and underscores oxidative stress as a core contributor to nanotoxicity.

While some safety concerns—such as ROS generation—are common across many nanodrug types, others are material-specific [6,10,35,120,122,136,138,176,179,184]. For example, carbon nanotubes have been associated with asbestos-like effects, while zinc oxide nanoparticles present toxicity related to their dissolution and ion release [246]. Among the most commonly reported strategies to mitigate these risks are surface modifications, which improve biocompatibility and reduce immune recognition (Table 5). Controlling physical parameters such as size and shape also emerged as critical to minimizing toxicity. These insights highlight the complexity of nanodrug safety and emphasize the need for comprehensive, material-specific approaches to toxicological assessment and risk management in nanomedicine development.

Overall, based on the structured assessment presented in this study, we have built a preliminary, concise, and structured checklist that aims to decrease nanodrug-induced toxicity (see Appendix A: *Cytotoxicity Checklist*). Our approach was inspired by the MIRIBEL guidelines (Minimum Information Reporting in Bio-Nano Experimental Literature), on which we propose laying the foundations to report experimental studies on bio–nano interactions [249]. Building on these thoroughly established guidelines, we provisionally propose a supplementary checklist aimed at reducing cytotoxicity specifically (see Appendix A: *Cytotoxicity Checklist*). Even though we are aware of the shortcomings of our checklist—most prominently its insufficiency of specificity regarding the exact physicochemical characteristics of NPs—we believe that even a preliminary structure of this kind constitutes an innovative input, especially given that, as far as we can tell, such a checklist has not yet been offered in the literature.

This checklist serves as a systematic methodology for minimizing the toxicity of the nanodrug without compromising therapeutic efficacy, thus allowing the construction of more secure nanopharmaceuticals for clinical use. The underlying principles of the checklist are: prioritizing active targeting over passive targeting to optimize selectivity; physicochemical optimization (e.g., particle size > 15 nm, neutral or negative charge, spherical shape); application of surface modification technologies such as PEGylation or biodegradable coatings; avoidance of key mechanisms of toxicity such as oxidative stress, inflammation, and lysosomal dysfunction; application of evidence-based risk assessment such as a Toxicity Index Scoring System (Figure 7); safety validation through rigorous mechanistic studies; incorporation of material-specific risks into the design phase. We hope this proposed checklist serves as a starting point for further refinement and validation within the scientific community.

Looking ahead, novel nanomaterials are being developed to enhance therapeutic performance while reducing adverse effects. Notably, advances in molecular self-assembly and nucleic acid nanotechnology have introduced new possibilities for precision nanomedicine [250,251,252,253,254,255,256]. Programmable biopolymeric materials (such as DNA nanocarriers) are particularly promising due to their inherent biocompatibility, structural precision, and molecular programmability [250,251,252,253,254,255,257]. DNA nanostructures (DNs) provide a versatile platform for nanoscale drug delivery, enabling controlled cargo release, improved targeting, and reduced systemic toxicity [36,254,258,259]. Although still in early stages of clinical development, DNA nanocarriers have demonstrated remarkable potential in enhancing drug delivery mechanisms and interacting with biological systems [36,254,258]. For example, functionalized DNA origami have been instrumental in dissecting nanoscale processes such as immune receptor activation [260,261,262,263], enzymatic activity [264,265], and spatial tolerances in antibody–antigen interactions [266]. Furthermore, DNA nanotechnology offers synthesis of highly versatile platforms for the precise modulation of cellular functions by targeting and regulating specific organelles through rational design [259,267,268,269,270]. This innovative approach represents a paradigm shift in nanomedicine, introducing a new class of nanocarriers with the potential to transform therapeutic strategies and significantly broaden the scope of biomedical interventions [271].

Overall, our analysis emphasizes that nanodrug-induced cytotoxicity results from a multifaceted interaction of cellular pathways, particularly those involving mitochondrial function, lysosomal integrity, ROS production, inflammation and membrane dynamics. By identifying common molecular mechanisms and emerging mitigation strategies, this work lays the groundwork for designing safer, more effective nanomedicines tailored to both clinical needs and biomedical applications.

## Figures and Tables

**Figure 1 ijms-26-06687-f001:**
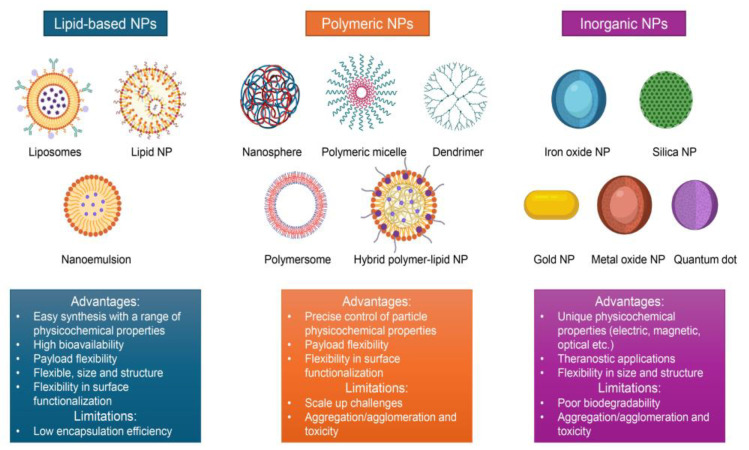
Schematic illustration of major nanoparticle types used in clinically approved nanodrugs. Each nanoparticle type includes several subtypes, a selection of which is highlighted here. These nanoparticle systems exhibit unique advantages and limitations related to cargo loading capacity, delivery efficiency, and patient-specific response profiles. Created with BioRender version 2.0.

**Figure 2 ijms-26-06687-f002:**
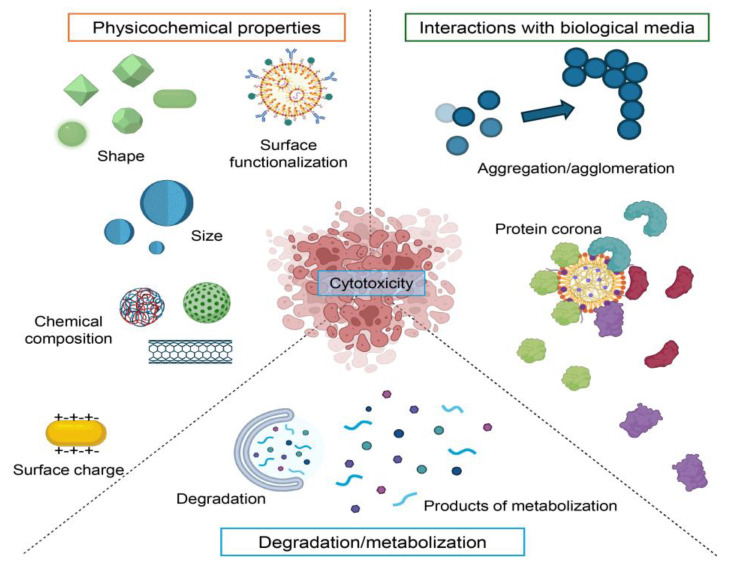
Summary of factors affecting nanoparticle cytotoxicity and biocompatibility. Created with BioRender.

**Figure 3 ijms-26-06687-f003:**
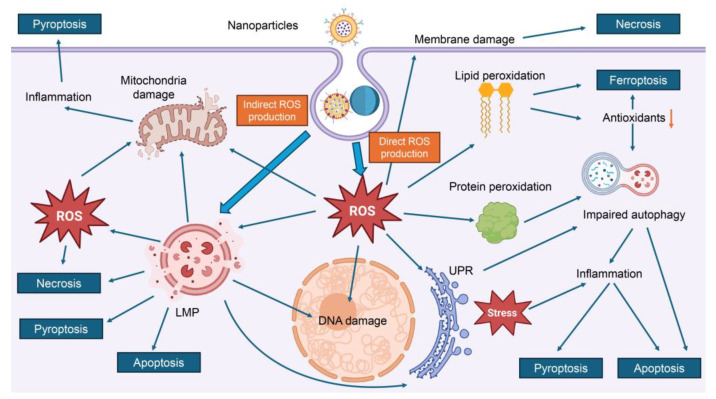
Schematic illustration of ROS role in cytotoxicity triggered by nanoparticles. Several factors contribute to the generation of ROS induced by nanoparticles, including their active surface area, size, photoactivation properties, associated toxins, metal ion dissolution, and interactions with biomolecules. ROS: reactive oxygen species; LMP: lysosomal membrane permeabilization; UPR: unfolded protein response. Created with BioRender.

**Figure 4 ijms-26-06687-f004:**
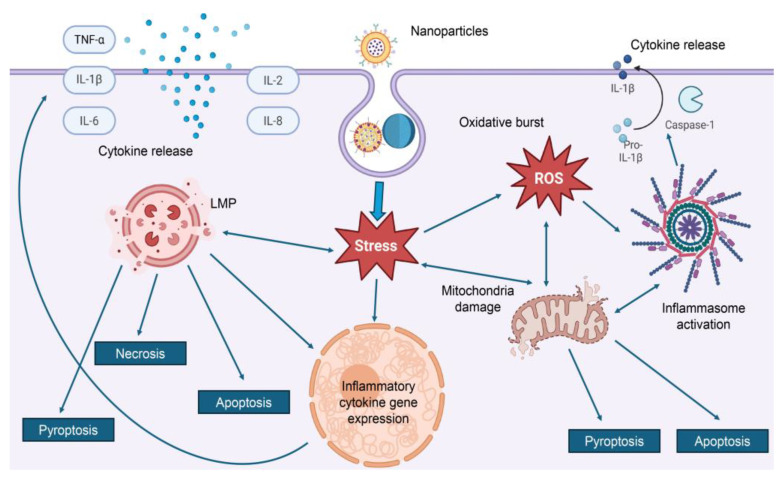
Role of inflammatory responses in cytotoxicity triggered by nanoparticles. ROS: reactive oxygen species; LMP: lysosomal membrane permeabilization; IL-6: Interleukin 6; IL-8: Interleukin 8; TNF-α: Tumor necrosis factor α; IL-1β: interleukin-1β. Created with BioRender.

**Figure 5 ijms-26-06687-f005:**
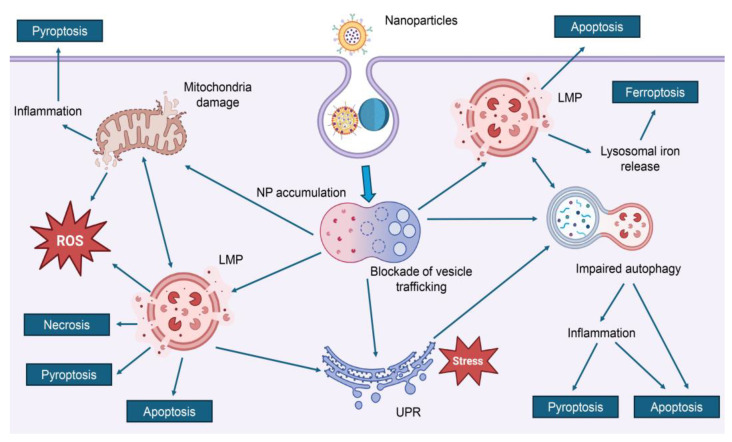
Role of lysosomal dysfunction in cytotoxicity triggered by nanoparticles. Nanoparticles may disrupt autophagy by overloading or damaging the lysosomal compartment, or by altering the cytoskeleton, thereby impairing autophagosome–lysosome fusion. Additionally, nanoparticles can compromise lysosomal stability by inducing oxidative stress, alkalization, osmotic swelling, or by exerting detergent-like effects on the lysosomal membrane, ultimately leading to LMP. ROS: reactive oxygen species; LMP: lysosomal membrane permeabilization; UPR: unfolded protein response. Created with BioRender.

**Figure 6 ijms-26-06687-f006:**
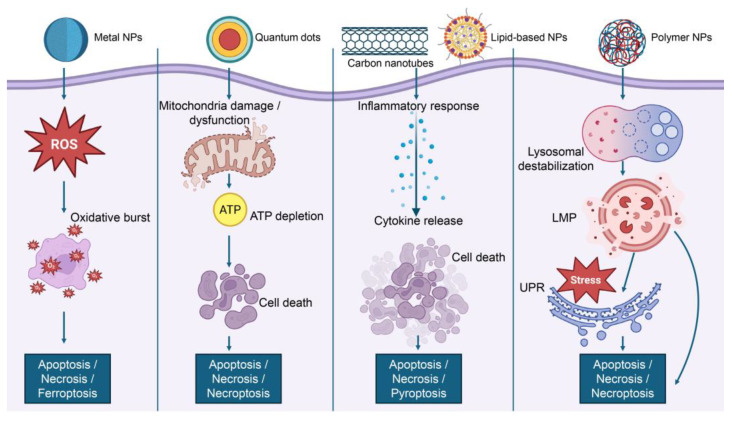
Generalized key cytotoxicity mechanisms associated with different classes of nanodrugs. ROS: reactive oxygen species; LMP: lysosomal membrane permeabilization; UPR: unfolded protein response. Created with BioRender.

**Figure 7 ijms-26-06687-f007:**
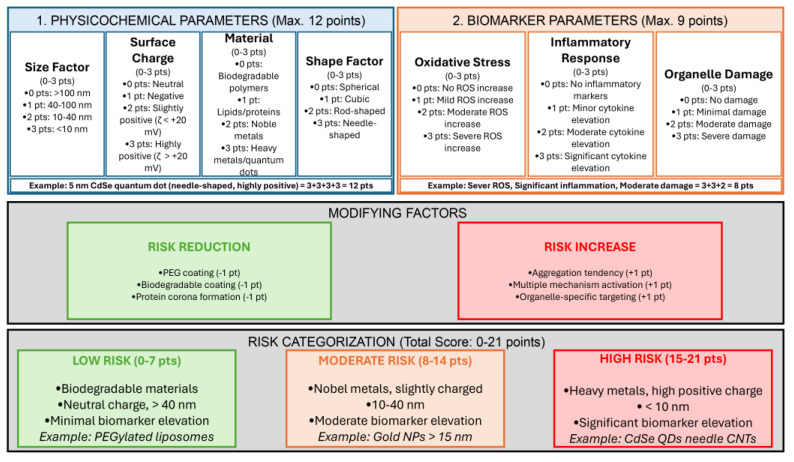
Proposed *Toxicity Index Scoring System*. QDs: quantum dots; NPs: nanoparticles; CNTs: carbon nanotubes; pt/pts: point/points. Created with BioRender.

**Table 1 ijms-26-06687-t001:** Clinically approved nanodrugs and their general characteristics.

Trade Name	API	Nanocarrier	Year of Approval	Applications	Company	Ref.
** *Lipid-Based NPs* **						
Diprivan	Propofol	Nanoemulsion	FDA 1989	Sedation or anesthesia	Fresenius Kabi	[9,48]
Doxil	Doxorubicin	PEGylated liposome	FDA 1995	Metastatic ovarian cancer HIV-associated Kaposi’s sarcoma Multiple myeloma	Janssen Pharmaceuticals	[9]
Abelcet	Amphotericin B	Lipid complex	FDA 1995	Invasive severe fungal infection	Leadiant Biosciences Inc.	[49]
Caelyx	Doxorubicin	PEGylated liposome	EMA 1996	Metastatic breast cancer Ovarian cancer AIDS-associated Kaposi’s sarcoma Multiple myeloma	Eagle Pharmaceuticals	[50]
AmBisome	Amphotericin B	Unilamellar liposome	FDA 1997	Fungal and/or protozoal infection Anti-leishmanial	Gilead Sciences, Inc.	[9]
Myocet	Doxorubicin	Liposome	EMA 2000	Metastatic breast cancer	CHEPLAPHARM Arzneimittel GmbH	[9,10]
Visudyne	Verteporfin	Unilamellar liposome	FDA 2000 EMA 2000	Decreased vision Macular degeneration Pathologic myopia	Bausch + Lomb	[51]
Definity	Perflutren	Phospholipid-stabilized microbubble	FDA 2001	Ultrasound contrast agent	Lantheus Medical Imaging	[52]
Mepact	Mifamurtide	Liposome	EMA 2009	Non-metastatic osteosarcoma and myosarcoma	Takeda Pharmaceuticals	[53]
Exparel	Bupivacaine	Liposome	FDA 2011 EMA 2020	Pain management	Pacira Biosciences	[54]
Onivyde	Irinotecan	Liposome	FDA 2015 EMA 2016	Metastatic breast cancer	Ipsen	[55]
Vyxeos	Daunorubicin and cytarabine (1:5 ratio)	Liposome	FDA 2017 EMA 2018	Acute myeloid leukemia AML-MRC t-AML	Jazz Pharmaceuticals	[56]
Onpattro	Patisiran sodium	Lipid nanoparticle	FDA 2018 EMA 2018	Hereditary transthyretin-mediated amyloidosis	Alnylam Pharmaceuticals, Inc.	[9,10]
Shingrix	Recombinant VZV glycoprotein E	Liposome	EMA 2018	Prevention of shingles and post-herpetic neuralgia	GlaxoSmithKline	[57]
Arikayce	Amikacin	Liposome	FDA 2018 EMA 2020	NTM lung disease caused by MAC	Insmed Incorporated	[58]
Comirnaty (BNT162b2)	mRNA encoding SARS-CoV-2 spike	PEGylated lipid nanoparticle	FDA 2021 EMA 2022	Prevention of coronavirus 2 infection (SARS-CoV-2 vaccine)	Pfizer, Inc.	[9]
Spikevax (mRNA-1273)	mRNA encoding SARS-CoV-2 spike	PEGylated lipid nanoparticle	FDA 2022 EMA 2022	Prevention of coronavirus 2 infection (SARS-CoV-2 vaccine)	Moderna, Inc.	[9]
** *Polymer-based NPs* **						
Optison	Perflutren	Albumin-stabilized microbubble	FDA 1997 EMA 1998	Ultrasound contrast agent	GE Healthcare	[59]
Abraxane	Paclitaxel	Albumin-bound nanoparticle	FDA 2005, 2012, 2013 EMA 2008	Metastatic breast cancer Lung cancer Metastatic pancreatic adenocarcinoma	Eli Lilly Company	[60]
Zilretta	Triamcinolone acetonide	PLGA microsphere	FDA 2017	Knee osteoarthritis	Pacira Biosciences	[61]
Apealea	Paclitaxel	Micelle	FDA 2018	Ovarian cancer Peritoneal cancer Fallopian tube cancer	Oasmia Pharmaceutical	[62]
** *Inorganic NPs* **						
Venofer	Iron sucrose	Colloidal iron sucrose	FDA 2000	Iron deficiency in CKD	American Regent	[63]
Feraheme	Ferumoxytol	Dextran-based nanoparticle	FDA 2009	Iron deficiency in CKD	Covis Group S.a.r.l	[26]
Injectafer	Ferric carbocymaltose	Colloidal formulation	FDA 2013	Iron deficiency in anemia	Vifor Pharma	[26]
Hensify	Hafnium oxide	Inorganic nanoparticle	EMA 2019	Squamous cell carcinoma	Nanobiotix	[64]

API: Active Pharmaceutical Ingredient; FDA: U.S. Food and Drug Administration; EMA: European Medicines Agency; VZV: Varicella-Zoster Virus; SARS-CoV-2: Severe acute respiratory syndrome coronavirus 2; AML-MRC: Acute myeloid leukemia with myelodysplasia-related changes; *t*-AML: Therapy-related acute myeloid leukemia; MAC: Mycobacterium avium complex; NTM: Nontuberculous mycobacteria; CKD: Chronic kidney disease; PEG: Polyethylene glycol; PLGA: poly(lactic-co-glycolic) acid.

**Table 2 ijms-26-06687-t002:** Key molecular pathways mediating nanoparticle-induced cytotoxicity.

Molecular Pathway	Affected Cell Components	Key Mediators	Clinical Implications	Ref.
Oxidative stress	Mitochondria, DNA, proteins, lipids	ROS, antioxidant enzymes	Potential for widespread cellular damage, mutagenesis	[179]
Inflammatory response	Cell membrane, cytokine signaling pathways	NF-κB, TNF-α, IL-6, IL-8	Chronic inflammation, tissue damage	[179]
Lysosomal dysfunction	Lysosomes, autophagy machinery	Cathepsins, autophagy-related proteins	Disruption of cellular waste management, potential trigger for cell death	[10,180,181]
Membrane disruption	Plasma membrane, organelle membranes	Membrane lipids, membrane proteins	Altered cellular permeability, potential for cell lysis	[182]
Mitochondrial dysfunction	Mitochondria	Electron transport chain components, ATP synthase	Energy metabolism disruption, potential trigger for apoptosis	[181]
DNA damage	Nucleus, mitochondrial DNA	DNA repair enzymes, p53	Mutagenesis, potential carcinogenesis	[179]
Apoptosis	Whole cell	Caspases, Bcl-2 family proteins	Programmed cell death, potential for tissue damage	[181,182]
Autophagy	Autophagosomes, lysosomes	LC3, p62, Beclin-1	Altered cellular homeostasis, potential protective or destructive effects	[10,181]
Ferroptosis	Cell membrane, mitochondria	Iron, lipid peroxides	Iron-dependent cell death, potential for tissue-specific effects	[183]
ER Stress	ER	Unfolded protein response proteins	Protein folding disruption, potential trigger for apoptosis	[181]
Cytoskeleton disruption	Actin filaments, microtubules	Actin, tubulin	Altered cell morphology and motility	[184]
Cell cycle regulation	Nucleus, cytoplasm	Cyclins, CDKs	Altered cell proliferation, potential for carcinogenesis	[184,185]
Immune modulation	Immune cells, cytokine signaling pathways	Toll-like receptors, complement proteins	Altered immune responses, potential for immunotoxicity	[186]
Epigenetic changes	Nucleus, chromatin	Histone modifying enzymes, DNA	Long-term alterations in gene expression	[187]

ROS: Reactive Oxygen Species; ER: Endoplasmic Reticulum; CDKs: cyclin-dependent kinases; NF-κB: nuclear factor kappa-light-chain-enhancer of activated B cells; IL-6: interleukin 6; IL-8: interleukin 8; TNF-α: tumor necrosis factor α.

**Table 3 ijms-26-06687-t003:** Molecular basis of nanoparticle cytotoxicity by particle type.

Nanoparticle Type	Molecular Pathways	Cell Model	Ref.
Gold nanoparticles	Oxidative stress Membrane disruption DNA damage ER stress Cell cycle regulation	Embryonic lung fibroblasts * Human neuroblastoma cells * Human monocytes * Human neutrophils * Human endothelial cells *	[181,182,183,213]
Silver nanoparticles	Oxidative stress Lysosomal dysfunction Mitochondrial dysfunction DNA damage Apoptosis Autophagy ER stress	Human leukemia cells * Human HCC * *D. melanogaster* ** AML cell lines * Embryonic lung fibroblasts * Human hepatoblastoma cells * AML patient samples ** Human bronchial epithelial cells * Mouse (lung, liver, kidney) ** Rat ** Human neuroblastoma cells *	[181,213,214,215,216,217]
Quantum dots	Oxidative stress Mitochondrial dysfunction Apoptosis ER stress Epigenetic changes	Human breast carcinoma * Endometrial cancer cells * Rat neuronal cells * Mouse ** Rat neuronal cells *	[181,182,187,218]
Carbon nanotubes	Oxidative stress Inflammatory responses Lysosomal dysfunction Mitochondrial dysfunction DNA damage Apoptosis	Rat embryonic lung fibroblasts * Normal and malignant mesothelial cells * Mouse and rat lungs ** Mouse hepatoblastoma cells * Lung epithelial cells * Human embryonic kidney cells * Bronchial epithelial cells *	[181,219,220,221,222,223,224]
Iron oxide nanoparticles	Oxidative stress Mitochondrial dysfunction DNA damage Apoptosis Cytoskeleton dysregulation Cell cycle regulation	Human breast carcinoma * Human cervical carcinoma * Human lung adenocarcinoma * Human embryonic kidney cells * Human neuroblastoma cells * Mouse hepatocytes * Breast cells (cancerous and non-cancerous) * Healthy lung cells * Liver cancer cells * Mouse ** Umbilical vein endothelial cells *	[99,183,184,225,226]
Silica nanoparticles	Oxidative stress Inflammatory responses Mitochondrial dysfunction Apoptosis Autophagy	Mouse macrophage-like cells * HUVEC * Mice (peritoneal macrophages) **, Mouse macrophage-like cells * Human endothelial cells * Human hepatic cell line * Human HCC *	[181,227,228,229]
Titanium dioxide nanoparticles	Oxidative stress Inflammatory responses Mitochondrial dysfunction Apoptosis Autophagy ER stress	Mouse erythrocytes, brain, liver ** Bronchial epithelial cells * Rat liver and kidney ** Rat ** Human monocytes * Human neuroblastoma cells * Mouse **	[181,183,230,231,232]
Zinc oxide nanoparticles	Oxidative stress Inflammatory responses Apoptosis Autophagy Ferroptosis	Mouse erythrocytes, brain, liver ** Rat lung **, Human lung adenocarcinoma * Human neuroblastoma cells * Mouse macrophages * Bronchial epithelial cells * Zebrafish embryos ** Embryonic lung fibroblasts * Mouse **	[181,182,183,230,233,234,235]
Aluminum oxide nanoparticles	Oxidative stress Autophagy Ferroptosis	Mouse erythrocytes, brain, liver ** Mouse ** Rat **	[181,183,230]

* in vitro model; ** in vivo model; HCC: Hepatocellular carcinoma; AML: Acute myeloid leukemia; HUVEC: Human umbilical vein endothelial cells.

**Table 4 ijms-26-06687-t004:** Qualitative comparative matrix showing the relationship between nanodrug types and their predominant cytotoxic mechanisms.

Nanodrug Category	Cytotoxic Trigger					Ref.
	Oxidative Stress	Lysosomal Dysfunction	Membrane Disruption	Inflammatory Response	Mitochondrial Dysfunction	
Liposomes	++	+	+++	++	+	[186]
Metal NPs (Au, Ag)	+++	++	+++	++	++	[182]
Quantum dots	+++	++	+	++	+++	[187]
Carbon nanotubes	++	+	++	+++	++	[116]
Polymeric NPs	+	+++	++	+	+	[180]
Iron oxide NPs	+++	+	++	++	++	[184]
Silica NPs	++	++	+++	+	+	[138]

+++ Strong association (frequently reported/primary mechanism); ++ Moderate association (commonly reported); + Weak association (occasionally reported).

**Table 5 ijms-26-06687-t005:** Risk mitigation strategies to counteract nanoparticle-induced cytotoxicity.

Nanodrug Category	Safety Concerns	Molecular Basis	Mitigation Approaches	Ref.
Metal nanoparticles (e.g., gold, silver)	Size-dependent toxicity, ROS generation	Oxidative stress, membrane disruption	Size optimization, surface coating (e.g., PEGylation)	[182,245]
Quantum dots	Heavy metal toxicity, long-term accumulation	Cadmium-induced cellular damage, ROS generation	Use of cadmium-free QDs, surface passivation	[187]
Carbon nanotubes	Asbestos-like effects, inflammatory response	Membrane damage, ROS generation, NF-κB activation	Functionalization, length control, use of biodegradable CNTs	[179,246]
Polymeric nanoparticles	Potential immunogenicity, complement activation	Protein corona formation, inflammatory response	Stealth coatings, immunomodulatory strategies	[180,186]
Liposomes	Complement activation, potential cardiotoxicity	Lipid peroxidation, membrane fusion	PEGylation, use of non-toxic lipids, size optimization	[186]
Iron oxide nanoparticles	ROS generation, potential for iron overload	Fenton reaction, disruption of iron homeostasis	Surface coating, controlled biodegradation	[184,186,241]
Silica nanoparticles	Hemolysis, liver toxicity	Membrane interactions, ROS generation	Surface modification, size control	[247]
Dendrimers	Cationic toxicity, hemolysis	Membrane disruption, mitochondrial dysfunction	Surface modification, use of biodegradable cores	[4,6,142]
Titanium dioxide nanoparticles	Pulmonary inflammation, potential carcinogenicity	ROS generation, DNA damage	Surface coating, shape control	[241,248]
Zinc oxide nanoparticles	Dissolution-related toxicity, ROS generation	Zn^2+^ release, oxidative stress	Surface passivation, controlled dissolution	[6,181]

ROS: Reactive Oxygen Species; NF-κB: Nuclear factor kappa-light-chain-enhancer of activated B cells; PEG: Polyethylene glycol; QDs: Quantum dots; CNTs: Carbon nanotubes.

## Data Availability

No new data were created or analyzed in this study.

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
