# Peer review of "Analyzing Molecular Determinants of Nanodrugs’ Cytotoxic Effects"

_ijms, 2025, doi:10.3390/ijms26146687_

Round 1

Reviewer 1 Report

Comments and Suggestions for Authors

Review of the article titled "Analyzing Molecular Determinants of Nanodrugs’ Cytotoxic Effects"

This article provides a thorough and well-structured review of the molecular mechanisms responsible for nanodrug-induced cytotoxicity. Using an AI-assisted literature screening approach, the authors effectively identify and analyze three key interconnected pathways—oxidative stress, inflammatory signaling, and lysosomal disruption—shedding light on how specific nanomaterial characteristics influence biological outcomes.

The work stands out for its wide scope, covering various nanodrug platforms from clinically approved systems to emerging technologies like DNA nanocarriers. The mechanistic detail is balanced with practical insights, particularly the inclusion of risk mitigation strategies linked to specific nanoparticle types. This integration of foundational science with applied considerations adds significant value for both researchers and practitioners in nanomedicine.

The "Outlook and Perspectives" section is particularly forward-looking, offering a thoughtful discussion on novel materials and future directions in safer nanodrug design.

In summary, this review offers an insightful and methodologically sound contribution to the field of nanotoxicology. I recommend accepting the article for further processing, as it provides a valuable foundation for designing safer, more effective nanotherapeutics.

Author Response

We sincerely thank the Reviewers for their constructive feedback. We have carefully considered all comments and made thorough revisions to the manuscript, which we believe have significantly enhanced its quality.

Below, we provide a detailed point-by-point response to each of the Reviewers’ remarks, along with the corresponding amendments made to the manuscript.

Answers to Reviewers’ questions and comments are given in blue and the changes in the revised manuscript are marked in red.

Comment:

We sincerely thank the Reviewer for their thorough and encouraging evaluation. We are pleased that the marriage of mechanistic insight with problems of practice was appreciated, and we are especially grateful for the generous praise of the range and structure of the review and the "Outlook and Perspectives" section. We believe that these aspects are crucial to advancing safer nanodrug design and are encouraged by the suggestion of the reviewer.

Reviewer 2 Report

Comments and Suggestions for Authors

The manuscript titled “Analyzing Molecular Determinants of Nanodrugs’ Cytotoxic Effects” addresses an important and clinically relevant challenge in nanomedicine—namely, understanding and mitigating nanodrug-induced cytotoxicity. The integration of AI-assisted literature screening and the focus on mechanistic toxicity pathways are timely and potentially impactful. However, the manuscript currently suffers from several substantive weaknesses that limit its novelty, analytical depth, and utility to the field. I recommend major revision before it can be considered for publication.

1.The identified toxicity pathways—oxidative stress, inflammation, and lysosomal disruption—are well-known and have been repeatedly discussed in earlier reviews. The current manuscript does not sufficiently differentiate itself by offering new mechanistic models, comparative analyses, or novel correlations between nanoparticle properties and biological outcomes.

2.The use of Elicit AI and other AI-assisted tools is briefly mentioned but inadequately explained. The authors should clearly describe the criteria for literature inclusion, the scope and limits of AI involvement, and how the AI results were validated or filtered by human judgment. Without this, reproducibility and objectivity remain in question.

3.The review appears to be largely descriptive. To improve its rigor, the authors are strongly encouraged to include quantitative data (e.g., frequency of mechanisms reported, material-specific trends, nanoparticle types and sizes most associated with specific pathways). A table summarizing the top studies reviewed—indicating particle type, cell model, mechanism observed—would be highly beneficial.

4.While the review suggests universal toxicity mechanisms, nanomaterials are highly diverse. The current form lacks proper stratification by nanomaterial class (e.g., metallic, polymeric, lipid-based), route of administration, or biological context. Specificity is critical to inform rational nanodrug design.

5.The claim that the review offers “indispensable guidelines” is not adequately supported by specific or actionable formulation strategies. Consider adding a dedicated section that translates your findings into concrete design principles—perhaps in the form of a decision tree or design checklist for minimizing cytotoxicity.

6.The manuscript presents AI as a strength but does not convincingly show how AI improved the comprehensiveness, quality, or objectivity of the review. Consider including a brief performance comparison between AI-assisted and traditional literature review workflows, or a figure showing how AI helped cluster mechanisms or materials.

Author Response

We sincerely thank the Reviewers for their constructive feedback. We have carefully considered all comments and made thorough revisions to the manuscript, which we believe have significantly enhanced its quality.

Below, we provide a detailed point-by-point response to each of the Reviewers’ remarks, along with the corresponding amendments made to the manuscript.

Answers to Reviewers’ questions and comments are given in blue and the changes in the revised manuscript are marked in red.

The manuscript titled “Analyzing Molecular Determinants of Nanodrugs’ Cytotoxic Effects” addresses an important and clinically relevant challenge in nanomedicine—namely, understanding and mitigating nanodrug-induced cytotoxicity. The integration of AI-assisted literature screening and the focus on mechanistic toxicity pathways are timely and potentially impactful. However, the manuscript currently suffers from several substantive weaknesses that limit its novelty, analytical depth, and utility to the field. I recommend major revision before it can be considered for publication.

Comment:

We appreciate the Reviewer's thoughtful comments and insights and thank them for recognizing the importance and validity of our work.

1.The identified toxicity pathways—oxidative stress, inflammation, and lysosomal disruption—are well-known and have been repeatedly discussed in earlier reviews. The current manuscript does not sufficiently differentiate itself by offering new mechanistic models, comparative analyses, or novel correlations between nanoparticle properties and biological outcomes.

Comment:

We appreciate this thoughtful remark of the Reviewer. While earlier reviews have concluded that oxidative stress, inflammation, and lysosomal injury are central nanotoxicity mechanisms, a thorough comparison of how clinically approved nanodrug formulations affect these mechanisms has been lacking. In this work, we aim to fill this void by linking certain nanocarrier properties with molecular cytotoxicity mechanisms and thereby establishing formulation-dependent risk profiles and therapeutic windows.

In order to more fully describe our findings and to respond to the recommendation of the Reviewer, Section 5 has been thoroughly rewritten. It now has two sections, with a new subsection entitled "5.2. Predictive Frameworks Determining Cytotoxicity of Nanodrugs" added to the manuscript.

Additionally, we included two new figures #6 and 7 and new table #3.

2.The use of Elicit AI and other AI-assisted tools is briefly mentioned but inadequately explained. The authors should clearly describe the criteria for literature inclusion, the scope and limits of AI involvement, and how the AI results were validated or filtered by human judgment. Without this, reproducibility and objectivity remain in question.

Comment:

We thank the Reviewer for this suggestion to enhance clarity and have included elaborated and contextualized discussion of AI usage to the revised manuscript.

We added the following discussion in the revised text.

Pages #3-4:

In order to systematically assess the current literature on nanodrug-induced cytotoxicity, we employed a multi-step search strategy combining traditional database searching with AI-driven tools, as well as manual reference list screening. The principal literature search was conducted using the Scopus and PubMed databases. To achieve greater coverage, we manually screened eligible articles' reference lists, thus allowing us to include supplementary relevant studies that could have otherwise been missed in light of varying keyword indexing. To supplement the depth and efficiency of our review, we incorporated Elicit AI (https://elicit.com), an artificial intelligence–based research assistant built on top of the Semantic Scholar corpus (https://www.semanticscholar.org/), which includes over 126 million scholarly publications. Elicit AI was specifically programmed to identify re-search on nanodrug-induced cytotoxicity. This yielded 498 papers initially, which were subsequently screened based on predetermined inclusion criteria. A complete list of these studies is given in the Supplementary Materials (Table S1).

The entire selection, screening, and data extraction workflow is summarized in Supplementary Materials (Figure S1), which presents a flowchart constructed in accordance with systematic reviews guidelines. The figure distinctly describes each phase of the re-view process—ranging from initial identification of records to final inclusion—marking methodological rigor and reproducibility. We applied the following inclusion criteria to screen articles retrieved by Elicit AI:

  1. Investigated FDA-approved or clinically used nanodrugs in therapeutic uses.
  2. Addressed molecular mechanisms of cytotoxicity induced by nanodrugs (e.g., protein corona formation, oxidative stress, inflammation, lysosomal damage).
  3. Provided experimental evidence (in vitro or in vivo) and not solely theoretical or computational.
  4. Provided a comprehensive toxicity or safety profile, such as dose-response curves or ICâ‚…â‚€ values.
  5. Analyzed nanodrug metabolism, degradation, or biodistribution in biological systems.
  6. Focused on therapeutic applications, excluding environmental or non-medical studies.
  7. Were original research articles or systematic reviews/meta-analyses.
  8. Addressed molecular-level toxicity mechanisms, beyond drug delivery efficacy

Each publication was evaluated holistically against these criteria, with inclusion decisions based on overall relevance and scientific merit. To prioritize studies for in-depth analysis, Elicit AI was used to assign a relevance score to each publication. These scores, presented in Supplementary Materials (Table S2), reflect how well each study aligned with the selection criteria. Applying a cutoff score of ≥3.5, we identified 98 high-priority articles for detailed evaluation.

For all these 98 studies, Elicit AI facilitated the quantitative and qualitative data ex-traction on nanomaterial cytotoxicity. Data extracted were:

  1. Toxicity rates or ICâ‚…â‚€ values.
  2. Dose-dependent toxicity responses.
  3. Molecular mechanisms (e.g., oxidative stress, inflammatory signaling, lysosomal disruption).
  4. Affected cellular structures (e.g., mitochondria, lysosomes, membrane integrity).
  5. Biomarkers of oxidative stress and inflammation.
  6. Observed immunomodulatory effects.

In reports describing several pathways of toxicity, Elicit AI was instructed to order mechanisms by reported rank or frequency of citation. If no explicit molecular mechanisms were reported, the report was labeled as "No specific molecular mechanisms re-ported."

To ensure consistent and targeted data extraction, Elicit AI was guided using the following screening questions:

Therapeutic Nanodrug Focus: Does the study investigate FDA-approved or clinically used nanodrugs and their cytotoxic effects at the molecular level?

Molecular Interaction Analysis: Are molecular-scale interactions (e.g., protein corona formation) between nanomaterials and cells examined?

Experimental Validation: Is the research based on empirical (in vivo or in vitro) data rather than computational or theoretical models?

Safety Assessment: Is a comprehensive toxicity or safety profile provided?

Biological Fate: Does the study examine nanodrug degradation, metabolism, or bio-distribution in biological systems?

Study Type Relevance: Is the context therapeutic (medical) rather than environmental or non-clinical?

Evidence Synthesis: Is the publication an original research article or a systematic re-view/meta-analysis?

Mechanism Analysis: Does the study focus on molecular mechanisms of cytotoxicity rather than drug delivery efficiency alone?

This structured, AI-assisted methodology ensured a robust, unbiased, and reproducible selection and analysis of literature addressing nanodrug-induced cytotoxicity.

3.The review appears to be largely descriptive. To improve its rigor, the authors are strongly encouraged to include quantitative data (e.g., frequency of mechanisms reported, material-specific trends, nanoparticle types and sizes most associated with specific pathways). A table summarizing the top studies reviewed—indicating particle type, cell model, mechanism observed—would be highly beneficial.

Comment:

We appreciate the Reviewer's comments and insights and included requested table in the revised manuscript. New Table 3 was added to the revised manuscript.

  1. While the review suggests universal toxicity mechanisms, nanomaterials are highly diverse. The current form lacks proper stratification by nanomaterial class (e.g., metallic, polymeric, lipid-based), route of administration, or biological context. Specificity is critical to inform rational nanodrug design.

Comment:

We appreciate the Reviewer's helpful comments and their recommendations, and we have added their requested discussion in the revised manuscript. We have elaborated particularly on characterization and generalization of cytotoxicity to other types of nanomaterials’ classes and justified this through the addition of a new figure (Figure 6).

In order to more accurately represent our findings and address the Reviewer's comments, Section 5 has been rewritten entirely. It is now divided into two subsections, with a new subsection entitled "5.2. Predictive Frameworks Determining Cytotoxicity of Nanodrugs."

We have also inserted two additional figures (Figures 6 and 7) and an additional table (Table 3) to further enhance and clarify the new material.

5.The claim that the review offers “indispensable guidelines” is not adequately supported by specific or actionable formulation strategies. Consider adding a dedicated section that translates your findings into concrete design principles—perhaps in the form of a decision tree or design checklist for minimizing cytotoxicity.

Comment:

We sincerely appreciate this valuable recommendation. We agree that phase “indispensable guidelines” is too strong, thus we replaced it. Further, we included discussion and provided tentative preliminary cytotoxicity scoring system. We also developed as suggested checklist for minimizing cytotoxicity.

We added the following discussion in the revised text.

Page #29:

Overall, based on the structured assessment presented in this study, we have built a preliminary, concise, and structured checklist that aims to decrease nanodrug-induced toxicity (see Supplementary Materials: Cytotoxicity Checklist). Our approach was inspired by the MIRIBEL guidelines (Minimum Information Reporting in Bio-Nano Experimental Literature), on which we propose laying the foundations to report experimental studies on bio–nano interactions [249]. Building on these thoroughly established guidelines, we pro-visionally propose a supplementary checklist aimed at reducing cytotoxicity specifically (see Supplementary Materials: Cytotoxicity Checklist). Even though we are aware of the shortcomings of our checklist – most prominently its insufficiency of specificity regarding the exact physicochemical characteristics of NPs – we believe that even a preliminary structure of this kind constitutes an innovative input, especially given that, as far as we can tell, such a checklist has not yet been offered in the literature.

This checklist serves as a systematic methodology for minimizing the toxicity of the nanodrug without compromising therapeutic efficacy, thus allowing the construction of more secure nanopharmaceuticals for clinical use. The underlying principles of the check-list are: prioritizing active targeting over passive targeting to optimize selectivity; physi-cochemical optimization (e.g., particle size > 15 nm, neutral or negative charge, spherical shape); application of surface modification technologies such as PEGylation or biode-gradable coatings; avoidance of key mechanisms of toxicity such as oxidative stress, in-flammation, and lysosomal dysfunction; application of evidence-based risk assessment such as a Toxicity Index Scoring System (Figure 7); safety validation through rigorous mechanistic studies; incorporation of material-specific risks into the design phase. We hope this proposed checklist serves as a starting point for further refinement and valida-tion within the scientific community.

6.The manuscript presents AI as a strength but does not convincingly show how AI improved the comprehensiveness, quality, or objectivity of the review. Consider including a brief performance comparison between AI-assisted and traditional literature review workflows, or a figure showing how AI helped cluster mechanisms or materials.

Comment:

We appreciate the Reviewer's helpful comments and suggestions. To this end, we included a new figure (Supplementary Figure S1) in the revised manuscript illustrating the workflow of literature search aided by AI. In addition, we have significantly revised the Methods section to provide more detailed description of the applied AI tools and to detail comprehensively the data extraction procedure from the publications in question.

Round 2

Reviewer 2 Report

Comments and Suggestions for Authors

Authors addressed all my concerns.